# The Healthy and Diseased Retina Seen through Neuron–Glia Interactions

**DOI:** 10.3390/ijms25021120

**Published:** 2024-01-17

**Authors:** Matheus H. Tempone, Vladimir P. Borges-Martins, Felipe César, Dio Pablo Alexandrino-Mattos, Camila S. de Figueiredo, Ícaro Raony, Aline Araujo dos Santos, Aline Teixeira Duarte-Silva, Mariana Santana Dias, Hércules Rezende Freitas, Elisabeth G. de Araújo, Victor Tulio Ribeiro-Resende, Marcelo Cossenza, Hilda P. Silva, Roberto P. de Carvalho, Ana L. M. Ventura, Karin C. Calaza, Mariana S. Silveira, Regina C. C. Kubrusly, Ricardo A. de Melo Reis

**Affiliations:** 1Laboratory of Neurochemistry, Institute of Biophysics Carlos Chagas Filho, Federal University of Rio de Janeiro, Rio de Janeiro 21949-000, Brazil; temponemh@biof.ufrj.br (M.H.T.); fggcesar@biof.ufrj.br (F.C.); dio.alexandrino@biof.ufrj.br (D.P.A.-M.); vtulio@biof.ufrj.br (V.T.R.-R.); 2Department of Physiology and Pharmacology, Biomedical Institute and Program of Neurosciences, Federal Fluminense University, Niterói 24020-150, Brazil; vladimirppbm@gmail.com (V.P.B.-M.); alinerabelo@id.uff.br (A.A.d.S.); mcossenza@gmail.com (M.C.); reginakubrusly1@gmail.com (R.C.C.K.); 3Department of Neurobiology and Program of Neurosciences, Institute of Biology, Federal Fluminense University, Niterói 24020-141, Brazil; camilasaggioro@id.uff.br (C.S.d.F.); alineteixeira@id.uff.br (A.T.D.-S.); egiestal@vm.uff.br (E.G.d.A.); robpaesuff@gmail.com (R.P.d.C.); almvuff@gmail.com (A.L.M.V.); kcalaza@id.uff.br (K.C.C.); 4Institute of Medical Biochemistry Leopoldo de Meis, Federal University of Rio de Janeiro, Rio de Janeiro 21941-902, Brazil; icaro.raony@bioqmed.ufrj.br (Í.R.); hercules.freitas@bioqmed.ufrj.br (H.R.F.); 5Laboratory of Gene Therapy and Viral Vectors, Institute of Biophysics Carlos Chagas Filho, Federal University of Rio de Janeiro, Rio de Janeiro 21949-000, Brazil; mdias@biof.ufrj.br (M.S.D.); hilda@biof.ufrj.br (H.P.S.); 6National Institute of Science and Technology on Neuroimmunomodulation—INCT-NIM, Oswaldo Cruz Institute, Oswaldo Cruz Foundation, Rio de Janeiro 21040-360, Brazil; 7Laboratory for Investigation in Neuroregeneration and Development, Institute of Biophysics Carlos Chagas Filho, Federal University of Rio de Janeiro, Rio de Janeiro 21949-000, Brazil; silveira@biof.ufrj.br

**Keywords:** retina, signaling, disease, neuron, glia

## Abstract

The retina is the sensory tissue responsible for the first stages of visual processing, with a conserved anatomy and functional architecture among vertebrates. To date, retinal eye diseases, such as diabetic retinopathy, age-related macular degeneration, retinitis pigmentosa, glaucoma, and others, affect nearly 170 million people worldwide, resulting in vision loss and blindness. To tackle retinal disorders, the developing retina has been explored as a versatile model to study intercellular signaling, as it presents a broad neurochemical repertoire that has been approached in the last decades in terms of signaling and diseases. Retina, dissociated and arranged as typical cultures, as mixed or neuron- and glia-enriched, and/or organized as neurospheres and/or as organoids, are valuable to understand both neuronal and glial compartments, which have contributed to revealing roles and mechanisms between transmitter systems as well as antioxidants, trophic factors, and extracellular matrix proteins. Overall, contributions in understanding neurogenesis, tissue development, differentiation, connectivity, plasticity, and cell death are widely described. A complete access to the genome of several vertebrates, as well as the recent transcriptome at the single cell level at different stages of development, also anticipates future advances in providing cues to target blinding diseases or retinal dysfunctions.

## 1. Introduction

The retina is the sensory tissue responsible for the first stages of visual processing. Retinal organization is as complex as other regions of the central nervous system (CNS), with quick and easy access and a wide neurochemical repertoire, such as in the brain. For these reasons, the retina is extensively used as a model for studying development and diseases [1,2,3,4] with several advantages, whether used in vivo, ex vivo (as explants), or in vitro. The retina can be dissociated to generate different cell cultures, such as: (i) mixed cultures; (ii) neuron-enriched cultures; (iii) purified as Müller glia cultures; and (iv) neurospheres [5,6]. Recently, functional platforms originated from stem cell organoids are being engineered to mitigate ocular diseases [7]. Neurogenesis and tissue development are widely described in the embryonic and mature retina [8]. Access to the complete genome of several vertebrates, including Mus musculus and *Gallus gallus*, as well as the transcriptome of individual cells at different stages of development, is available [9,10].

All this information provides useful tools to translate into experimental strategies. No less important, the cells that make up the retina express most of the transmitters and modulators present in other regions of the CNS. These advantages and the importance of the retina as a key sensory tissue, together with the fact that most diseases that cause blindness are consequences of retinal dysfunction, make the retina a fascinating model for the analysis of neural structure, function, development, and diseases [11].

## 2. The Organization of the Retina

The basic plan of the retina’s organization is a highly conserved structure among all vertebrates. Five types of neurons are organized in three cell layers (nuclear) separated by two layers of synaptic contacts (plexiform). The photoreceptors are in the outer nuclear layer (ONL); three types of interneurons (bipolar, amacrine, and horizontal cells) are in the inner nuclear layer (INL); and finally, the retinal ganglion cells and displaced amacrine cells are in the ganglion cell layer (GCL). Photoreceptors capture light and transform it into electrochemical signals. They make synapses with bipolar and horizontal cells in the outer plexiform layer (OPL), while ganglion cells make synapses with bipolar cells and amacrine cells in the inner plexiform layer (IPL). The axons of ganglion cells project from the eye and form the optic nerve, which will communicate with other brain regions to continue visual processing. Each class of cells are linked with specific connectivity patterns that generate ganglion cells with different sensitivities for stimuli, such as edges, color contrasts, and moving or stationary objects. In addition, the retina also contains glial cells, the major one being the Müller glia, which interact symbiotically with all layers of the retina, potentially communicating with all cellular types. The intimate interaction between Müller glia and retinal neurons and microglia through secreted factors, including neurotransmitters, has been investigated in physiological and pathophysiological contexts [12]. Moreover, Müller glia’s role as an endogenous regenerative cell source in teleost fish and as a potential target for the development of new regenerative approaches in mammals have also received attention [13]. In addition, astrocytes, which are mostly found in the nerve fiber layer and microglia, that invade the retinal tissue during the embryonic period are also key players in retinal homeostasis and diseases [14].

Despite the retina’s plan being preserved, the proportion and characteristics of cell types and subtypes and connectivity patterns vary among species. Birds are highly visual, with relatively large eyes compared to their skull, with sophisticated and high-acuity retinas [15]. For example, while most mammals have two types of cone photoreceptors, most birds are tetrachromatic, with their cones being sensitive to red, green, blue, and ultraviolet light [16].

The synapses between photoreceptors, bipolar, and ganglion cells are defined as the vertical pathway, and communication is mainly carried out through glutamatergic release in mature tissues. The communication between these cell types and horizontal and amacrine cells is defined as the horizontal pathway, and it mainly works through inhibitory (GABAergic) synapses. In addition, there is an inhibitory structure endorsed by the GABAergic (and/or glycinergic) system in the horizontal pathway, mediated by horizontal and amacrine cells, which modulate neuronal excitability and the vertical pathway, and is responsible for light preprocessing, such as contrast and approach sensitivity [17,18]. Although GABA and glutamate may be considered the main neurotransmitters, the vertebrate retina presents most of the known neurotransmitters and multiple neuropeptides [19].

Diverse messengers, such as acetylcholine, ATP, dopamine, adenosine, and serotonin, peptides, such as PACAP (pituitary adenylyl cyclase polypeptide) and VIP (vasoactive intestinal peptide), and lipid endocannabinoids (eCB) are present in the retina (Figure 1). The production and release of these molecules influence the functioning control of the cell cycle and neuronal differentiation throughout development to provide a mature retinal circuitry [11,20].

## 3. Neurotransmitters

### 3.1. Glutamate

Glutamatergic communication is essential to the retina, present early in development, and its dysfunctions are implicated in several disorders. Excitotoxicity, which occurs through the increased influx of calcium through ionotropic glutamate receptors (iGluRs), is often implicated in neuronal damage. Studies with cells or retinospheroids have shown that relatively short incubation periods (between 30 min and 2 h) are enough to induce neuronal death, usually observed between 8 and 24 h after exposure, with a few dying numbers after 48 h. Calcium influx through iGluRs correlates with cell death [21]. In this sense, the development of retinal degeneration is not only strongly implicated in excitotoxicity but also in inflammation and oxidative stress, such as underlying mechanisms in glaucoma, retinitis pigmentosa, diabetic retinopathy, and retinal ischemia, among other diseases [22,23]. These effects are largely explained by the increase in the extracellular levels of excitatory amino acids (EAAs), leading to retinal remodeling. Indeed, exposure to an EAA or its analogues, depending on their concentration and exposure time, leads to retinal cell death in two-dimensional cultures, organotypic cultures, and/or ex vivo models [21,24]. The intravitreal administration of NMDA, an agonist of a receptor subtype of glutamate, is used to induce retinal excitotoxicity [25,26]. The NMDA receptor is also involved in synaptic plasticity, memory, and learning, to name a few physiological tasks. Although not exclusively, it has been shown as an essential mediator involved in ischemia and cell death in the last decades [27,28].

Calcium mobilization is essential for a myriad of phenomena following the birth of a cell, including growth, proliferation, cytoskeletal remodeling, adhesion cells, and transcription of genes, but also in the activation of proteases such as caspases and in the induction of cell death [29]. Therefore, the availability of Ca^2+^ is highly regulated through the actions of proteins that regulate its levels, and cell imaging has been used as a reliable method to study neural–glial circuits in the last four decades [30]. Excessive calcium entry induces excitotoxicity, which leads to death through the activation of calcium-dependent enzymatic systems such as nitric oxide neuronal synthase (nNOS), calpains, and phospholipases [31]. An excessive disruption of calcium causes the disturbance and loss of mitochondrial potential (∆Ψ), activating death-programmed and unscheduled pathways [32]. In particular, the GluN2B subunit is linked to PSD-95 through its carboxyl terminus, which couples to nNOS, which, from the excessive activation of the receptor, produces pro-death signals, leading to a reduction in cyclic AMP (cAMP) responsive binding protein (CREB) activity [24,32,33]. Due to the rich presence of iGluRs, the retina tissue is extremely vulnerable to excitotoxicity, a mutual mechanism for several diseases, including hypoglycemia, hypoxia, ischemia, and chronic neurodegenerative diseases [34]. On the other hand, excessive activation of AMPA receptors has also been correlated to ischemia-like insults to retinal ganglion cells (RGCs) [35], and its control might be a relevant therapeutic target in ocular neuropathies [34].

NMDAR stimulation activates the enzyme calcium calmodulin kinase III (CaMKIII) [36,37], currently known as eukaryotic elongation factor 2 kinase (eEF2K), due to the recognition that its only activity is the phosphorylation of Thr-56 of the translation factor eEF2 [38,39,40]. eEF2 mediates the translocation of peptidyl-tRNA from the ribosomal A-site to P-site by GTP hydrolysis, consuming a significant amount of energy. When phosphorylated, this factor hinders the elongation phase of protein synthesis, blocking the growth of the polypeptide chain [41,42]. As a calcium–calmodulin complex (Ca^2+^–CaM)-dependent kinase, this enzyme has been implicated in several signaling processes, which require the rapid and transient inhibition of protein synthesis. Interestingly, over short timescales, changes in neuronal protein synthesis can occur completely independently of new transcription, for example, in response to stimuli in synaptosomes [43], isolated axons [44], and in dendritic spines [45].

It has been reported that the expression of some synaptic proteins, such as the alpha subunit of CaMKII in isolated synaptosomes [46] and brain-derived neurotrophic factor (BDNF) [47], paradoxically increases with NMDAR/Ca^2+^-CaM/eEF2K activation, despite this pathway inhibiting general protein synthesis, an effect that is not yet completely understood. The activation of the NMDAR/Ca^2+^-CaM/eEF2K pathway was described to enhance the availability of intracellular free L-arginine, contributing to increased NO synthesis by nNOS [36,37]. Therefore, NMDA-triggered Ca^2+^ signaling could operate in two different ways to increase NO production: (1) by activating nNOS directly; and (2) by supplying the nNOS substrate, L-Arg. According to these models, protein synthesis could play an active role in the regulation of L-Arg “pools” and the synthesis of NO in neurons.

Furthermore, eEF2 phosphorylation mediated by NMDAR activation was clearly associated with CREB activation in the retina, an event that appears to depend on the increase in free L-arginine and the activation of nNOS [37]. Increased L-arginine has also been described during pharmacological treatments with cycloheximide or anisomycin (CHX or ANISO, respectively), where the PKG-dependent NO signaling pathway (a canonical pathway) was shown to activate ERK and AKT [36,48]. It is known that CREB, AKT, and ERK are widely associated with promoting survival, neuronal growth, synaptic plasticity, response to stress, and learning and memory [49,50,51].

### 3.2. γ-Aminobutyric Acid (GABA)

GABA is widely reported as the main inhibitory transmitter of the mature CNS of vertebrates [52,53,54], being estimated that around one-third of all neurons are GABAergic [55]. GABA^+^ cells are mainly interneurons that are responsible for controlling the excitability of the local circuitry [56,57]. GABA is found in the retina of several vertebrates [53], present in subpopulations of horizontal, amacrine, and ganglion cells, and in Müller glia in the avian retina [8,58,59,60,61,62].

GABA is mainly synthesized from glutamate via glutamic acid decarboxylase (GAD) and stored in vesicles by the GABA vesicular transporter (VGAT) until its release at the synaptic cleft [63]. GAD has two isoforms named after their molecular weights, GAD65 and GAD67 [64], which are rate-limiting enzymes that maintain GABA levels [65]. GAD65 has been reported as a specialized and ready-to-synthesize GABA under short-term demand. It is mostly found on nerve terminals and has a readily inducible state, which depends on neuronal activity [66]. It has also been described as essential for neuroprotection and development [63,67]. GAD67 has a more dispersed distribution across GABAergic neurons while mostly fully activated [66] and may also be responsible for glial GABA synthesis [68]. Recently, it has also been described that glial GABA synthesis is generated by diamine oxidase (DAO) and aldehyde dehydrogenase A1 (Aldh1a1) [69]. In the enteric nervous system [70] and midbrain dopaminergic neurons, GABA is also suggested to be produced by putrescine via ornithine decarboxylase [71,72] and diamine oxidase (DAO) [73]. Interference with GABA synthesis and uptake is linked to several pathologies, such as schizophrenia [74]. Interestingly, retinal glia are highly involved in glutamate and GABA uptake in the retina, and Müller cells are affected by diabetes, turning into a reactive state and incapable of efficient antioxidant control [23,75].

GABA receptors are classified as GABA_A_, GABA_B_, and GABA_C_. GABA_A_ and GABA_C_ are ligand-gated ion channels that are permeably selective to Cl^−^, with GABAA also being permeable to bicarbonate (HCO_3_^−^) to a lesser extent; GABA_B_ is a G protein-coupled receptor [54,76].

After its release, GABA is cleared from the synaptic cleft by re-uptake through its high-affinity GABA transporters (GATs) in a Na^+^ and Cl^−^ symport with substrate-dependent and ligand-gated ion channel properties [77]. These transporters are present in the presynaptic terminals of GABAergic neurons and glial cells [78,79]. The GATs belong to the sodium symporter family, also known as the solute carrier 6 (SLC6) family [77,80], and are responsible for GABA uptake from the extracellular environment in favor for the Na^+^ gradient maintained by the Na^+^/K^+^ ATPase pump; however, it can also release GABA through a transport gradient reversal mechanism in the retina [81,82]. In mammals, but not in avian retina, Müller glia uptake and recycle GABA. Similar reversal mechanisms were described for other neurotransmitter transporters, such as for dopamine, glutamate, serotonin, and glycine [83,84,85].

GABA is released into the synaptic cleft in retina when stimulated by depolarization and exerts its effects pre- and post-synaptically via ionotropic (GABA_A_ and GABA_C_ receptors) and metabotropic (GABA_B_) receptors. In the retina, the activation of iGluRs in amacrine and horizontal cells promotes GABA release [86]. Dopamine inhibits the release of GABA induced by NMDA but not by kainate, whose effect could act directly in or near the NMDA receptor complex through mechanisms that seem not to involve known dopaminergic receptor systems [81,87,88]. Nitric oxide (NO), an endogenous mediator in the retina, might regulate GABA release in a biphasic manner. Low and moderate NO production inhibit basal GABA release, mainly from amacrine cells and ganglion cell layer (GCL) cells, while NMDA or L-arginine (at high concentration) induce a NO-dependent increase in GABA release in GCL cells [89]. The GABAergic system delineates an important physiological significance to modulate and contribute to the control of sensory inputs in retinal function.

In the chicken retina, GAT-1 is responsible for about 90% of GABA uptake [90,91]. This transport is dependent on Na^+^ and Cl^−^ and independent of Ca^2+^ [80,92], indicating that in the retina, GABA release is mainly mediated by receptor reversal and not exocytosis [59].

Additionally, in the chick retina model, it is known that several neurotransmitters and drugs might modulate the release of GABA, including glutamate, via NMDA and non-NMDA receptors [59,81,86,88,93,94], aspartate, via the selective activation of NMDA receptors [88,95], ethanol [96], dopamine [88], and adenosine receptors, via protein kinase C (PKC) [97] and via A_1_R blockade with caffeine [59,93].

### 3.3. Dopamine

Dopamine is known as one of the main mediators in the vertebrate retina present in amacrine cell bodies and processes [98]. The synthesis of dopamine, as well as the other catecholamines, depends on tyrosine hydroxylase (TH), the limiting enzyme for the synthesis of catecholamines and its cofactor tetrahydropterin, converting L-tyrosine into L-DOPA (3,4 dihydroxy-phenylalanine). After L-DOPA synthesis, it is rapidly metabolized to dopamine by aromatic amino acid decarboxylase (AADC) or dopa decarboxylase (DDC), which are also capable of decarboxylating the amino acid tryptophan.

The change in the endogenous levels of dopamine may be correlated with disorders such as myopia since, during development, dopaminergic signaling regulates visual acuity, and its modulation depends on several factors such as visual stimuli or chemical mediators [99]. Changes in the dopaminergic system are correlated with several effects, such as modification of neurogenesis, reduction in the level of filopodial activity, neurite retraction, reduction in the conductance level of GAP junctions, inhibition of GABA release, reduction in the level of apoptosis, and regulation of spontaneous neural activity [100]. These functions mainly seem to be linked to D_1_ receptor-mediated effects [101].

Although noradrenaline and adrenaline also act as neurotransmitters in the mammalian retina, studies carried out on the chick retina demonstrate the absence of the noradrenaline-producing enzyme dopamine-beta-hydroxylase, characterizing an absence of noradrenaline and adrenaline synthesis in this model being present, therefore only the dopaminergic system [102]. Concomitantly, the retinal pigment epithelium (RPE) is capable of replenishing L-DOPA and synthesizing dopamine due to the expression of DDC [103]; however, it does not express the dopamine transporter (DAT), indicating the existence of another mechanism of dopamine transport through the membrane [104].

Dopamine receptors have been described in early stages of embryonic chick development [105] and has key roles in mature neurons. D_1_ receptors can be classified into the subtypes D_1A_ and D_1B_, which have different roles during differentiation [106,107]. A transient dopamine receptor controls the effects of dopamine on the morphology and motility of cultured retinal neurons [108]. Indeed, a transient β_1_ adrenergic receptor was also found in the avian retina through the detection of mRNA and β_1_ adrenergic receptor protein in post-hatched tissue [109]. It was shown that norepinephrine cross-reacts with D_1_ dopaminergic receptors like dopamine in the embryonic retina, but as the retina matures, selective D_1_ receptor activation by dopamine or β_1_-like adrenergic receptors occurs in the mature tissue [109].

Components of the dopaminergic system are detected throughout the differentiation of amacrine cells in an embryonic chick at E3–E8, functional DAT arises around E8, and the D1 receptor can be detected around E7. These structures appear before the first spontaneous electrical activity in the developing retina (E8–E11). Additionally, TH, which is one of the most important molecular components to characterize the dopaminergic phenotype, has been reported to appear later in development in the chicken retina (E12). After the maturation of the dopaminergic system, dopamine levels increase at around E15, which coincides with the peak of the A_1_ adenosine receptor’s density (E15) and a peak of intracellular cAMP accumulation (E16) by exogenous dopamine activation [8,100,103,109]. It is also known that during this period, a dopaminergic stimulus will promote an increase in cAMP and that stimulation with adenosine agonists will be able to partially inhibit this increase. The A_2_ adenosine receptor is expressed from E14, while A1 is present from E11 onwards [110]. The increase in cAMP levels is one of the factors underlying the increased differentiation of TH-positive cells [111]. Activation of PAC1 receptors by PACAP generates an increase in cAMP levels in chick retinal cultures and modulates the expression of TH-positive neurons [112].

Among the endogenous/exogenous factors specifically involved in the differentiation of dopaminergic cells are the drugs that increase cAMP levels [111].

### 3.4. The Endocannabinoid System

The endocannabinoid system controls neural excitability, mainly through the modulation of glutamate and GABA release, suggesting a relevant role in the process of visual encoding. In this sense, it has been identified as the main circuit breaker in the nervous system [113], known to be involved in the modulation of synaptic transmission and plasticity [114] and in several physiological processes, from embryogenesis to late development and homeostasis maintenance in the mature tissue [115]. It is commonly acknowledged as the most abundant synaptic system in the brain [116], present early in the development of neurons and glial cells [117]. This also happens in the retina, where several markers (receptor, enzymes, and transporters) have been functionally characterized [118,119] in addition to its messengers (anandamide, 2-aracdonoyil glycerol, and others), which are involved in visual processes [117,120,121] and under pathophysiological conditions affecting the ocular system, such as in glaucoma or diabetic retinopathy [122,123,124,125]. The expression of the cannabinoid receptors (e.g., CB_1_ and CB_2_) as well as transient receptor potential vanilloid (TRPV) channels in the vertebrate retina starts early during retinal development.

The presence of the endocannabinoid system in the retina began to be investigated from the end of the 1990s. Initially, Schlicker demonstrated that cannabinoid agonists were capable of inhibiting dopamine release in guinea pig retinas [126]. Buckley and coworkers observed that CB_1_ mRNA was detectable from E11 during rat embryogenesis [127]. Retinal development begins around E12 with ectoderm invagination in rats, which generates the lens vesicle and the inner neuronal layer of the optic cup (future neuronal layer of the retina) [128]. CB_1_ mRNA is detectable in the inner layer from E12, and in E13, it is already present in the retina, showing the importance of this system for development [127].

CB_1_ is highly conserved in the mature retina among vertebrates, as it was identified in rhesus monkeys, mice, rats, chicks, goldfish, and tiger salamanders, to name a few [129]. These receptors are generally located at the synaptic layers, the inner and outer plexiform layers, in cones and/or rods, amacrine cells, and ganglion cells [130]. Other elements of the system are also found in ocular tissue, such as the ligands and enzymes involved in the synthesis and degradation of endocannabinoids [131]. Functionally, it has been shown that cannabinoid agonists decrease the amplitude of voltage-gated L-type calcium channel currents in retinal bipolar cells, indicating their role in neuronal communication [130]. They also modulate calcium shifts in avian retinal Müller cells induced by ATP, but not in depolarized neurons [119]. Indeed, cannabinoid CB_1_ and purinergic P2X7 receptors have a role in avian retinal progenitors [132]. Additionally, some aspects of retinal processing, such as the modulation of response strength to visual stimulation, receptive field organization, and contrast sensitivity, are also modulated by tonic endocannabinoid release in the retina [133]. 

In the retina, CB_1_ is detected in ganglion cells from embryonic day 3 (~E3) [134]. Corroborating with their results, Leonelli and coworkers showed the presence of the CB_1_ receptor in the retinotectal system using conventional immunoperoxidase protocols. In their study, weak CB_1_ labeling was detected from E4 in the retina and optic tectum, with the signal raising over development [135]. The endocannabinoid system is classically composed of cannabinoid receptors, eCBs, and the enzymes responsible for their synthesis and degradation. It is known that there are two major types of receptors, CB_1_ and CB_2_, with both receptors coupled to a G protein and involved in several cell signaling systems [136]. The activation of CB_1_ and CB_2_ is classically followed by the reduction in the intracellular levels of cAMP, a consequence of the inhibition of the enzyme adenylate cyclase by the involvement of the Gi protein [137], among other elements.

Regarding function, Warrier and Wilson demonstrated that eCBs play a modulatory role in regulating the release of neurotransmitters from embryonic retinal amacrine cells, indicating their involvement in fine-tuning synaptic transmission during the developmental stages of the visual system [138]. Chaves and colleagues explored the consequences of retinal removal on the expression of cannabinoid CB_1_ receptors in the optic tectum of chick brains [139]. Adult chicks were used in experiments conducted at various time intervals post-retinal lesion (ranging from 2 to 30 days). Notably, this study revealed no evidence of cell death in the deafferented tectum within the first 30 days post-lesion, although Fluoro-jade B staining did indicate degenerating axons and terminals. Retinal ablation led to an increase in CB_1_ receptor protein levels in the optic tectum, as well as in other retinorecipient visual areas, coinciding with heightened MAP-2 staining and suggesting dendritic remodeling. However, CB_1_ receptor mRNA levels remained unaltered following retinal removal. These results imply that CB_1_ receptor expression in visual structures of the adult chick brain may be negatively regulated by retinal innervation. The increased CB_1_ receptor expression level observed after retinal removal suggests that these receptors are not presynaptic in retinal axons projecting to the tectum, pointing to a potential role of the cannabinoid system in plasticity processes ensuing after retinal lesions.

Cannabinoid receptors and transient receptor potential ankyrin (TRPA_1_) channels were also explored in the context of retinal ischemia, a condition marked by an inadequate blood flow to the retina, often associated with vision loss and a lack of effective treatments. The research by Araújo et al. explored the use of cannabinoid system modulation to mitigate cell death triggered by acute ischemia in an avascular (chick) retina [140]. A combination of WIN 55212-2 (a cannabinoid receptor agonist) and cannabinoid receptor antagonists (AM251/O-2050 or AM630) was shown to reduce the release of lactate dehydrogenase (LDH) induced by retinal ischemia in an oxygen and glucose deprivation (OGD) model. Surprisingly, administering any of these drugs individually did not prevent LDH release during OGD. This suggests that the increased availability of eCB combined with cannabinoid receptor antagonists has a neuroprotective effect in the context of retinal ischemia. This study also explored the involvement of TRPA1 receptors in retinal cell death during ischemic events. TRPA_1_ levels increased after OGD. Notably, the selective activation of TRPA_1_ did not worsen LDH release during OGD, while blocking TRPA_1_ completely prevented LDH leakage under ischemic conditions. This indicates that TRPA_1_ activation plays a critical role in inducing cell death during ischemia. This study suggests that metabotropic cannabinoid receptors, including type 1 and type 2, are not associated with cell death during the early stages of ischemia, pointing to the potential utility of targeting TRPA_1_ for neuroprotective strategies in the context of retinal ischemia. Overall, this research offers insights into the potential mechanisms underlying neuroprotection during retinal ischemia and identifies TRPA_1_ as a promising target for future neuroprotective interventions in this condition.

WIN 55,212-2 was also shown to decrease cAMP production in cultured avian embryonic retinal cells under basal conditions. WIN had an impact on glial cells, reducing the calcium levels evoked by ATP but not affecting calcium shifts in neuronal cells activated by KCl. Furthermore, WIN inhibited the GABA release induced by KCl or L-aspartate in amacrine cells but had no effect on GABA release under an OGD condition. This research underscores the crucial role of cannabinoid receptors in regulating signaling during synapse formation in the avian retina during critical embryonic stages, providing valuable insights into the expression and functions of CB_1_ and CB_2_ receptors in retinal cells, particularly their influence on cell excitability and GABA release [141,142,143]. In the avian retina, progenitor emergence around the first embryonic week is modulated through cannabinoid receptor activation (by the CB_1_/CB_2_ agonist WIN 5212-2 (WIN) [132,144]. In our hands, retinal cells in culture respond selectively to KCl and/or AMPA (neurons) or ATP (glia), while progenitor cells are activated by muscimol or GABA [132,145].

We have previously shown that chronic incubation of retinal cells in culture with WIN selectively decreases calcium response to ATP but not to KCl, suggesting that somehow glial cells, but not neurons, are modulated by cannabinoid receptor activation [119]. Therefore, in addition to regulating cAMP production, [^3^H]-GABA release induced by KCl or L-ASP, or [^3^H]-D-ASP release by KCl in cultured avian retinal cells [119], WIN also decreased the number of glial cells that responded with Ca^2+^ shift levels evoked by ATP but did not alter neuronal cells activated by KCl [119]. Therefore, cannabinoid receptors function as regulators of avian retina signaling at critical embryonic stages during synapse formation.

The cannabinoid agonist WIN 55,212-2 was also used to investigate the developmental properties of the retinal glial progenitor cells. The findings from Freitas et al. indicate that WIN treatment leads to a reduction in [^3^H]-thymidine incorporation and a decrease in the number of proliferating cell nuclear antigen-positive nuclei (PCAN^+^) counts, suggesting that the activation of cannabinoid receptors hampers the proliferation of cultured retinal progenitors [132]. Additionally, WIN treatment reduces retinal cell viability, an effect that can be blocked by CB_1_ and CB_2_ receptor antagonists, as well as the P2X7 receptor antagonist A438079. This implicates the P2X7 nucleotide receptor in cannabinoid-mediated cell death. Moreover, WIN induces an increase in mitochondrial superoxide and enhances the P2X7 receptor-mediated uptake of sulforhodamine B in cultured cells. While a substantial proportion of cultured cells respond to glutamate, GABA, and high levels of KCl with intracellular calcium shifts, only a few cells respond to the activation of P2X7 receptors by ATP. Remarkably, treatment with WIN decreases the number of cells responding to glutamate, GABA, and KCl, but significantly increases the number of cells responding to ATP, suggesting that the activation of cannabinoid receptors primes P2X7 receptor-mediated calcium signaling in retinal progenitors in culture.

Campbell and colleagues also investigated the involvement of the eCB system in the proliferation of progenitor-like cells in the retina [146]. Their research involved a comprehensive characterization of the expression patterns of eCB-related genes in both chick and mouse models. Their findings revealed that CNR1, the eCB receptor, and the enzymes related to eCB metabolism are expressed in the Müller glia (MG) and inner retinal neurons. In the chick model, intraocular injections of cannabinoids, specifically 2-arachidonoylglycerol (2-AG) and N-arachidonoylethanolamine (AEA, anandamide), were shown to stimulate the formation of MG-derived progenitor cells (MGPCs). This study also demonstrated that pharmacological agents targeting the eCB system can significantly influence glial reactivity and the capacity of MG to transition into MGPCs. Moreover, in damaged mouse retinas where the MG activate NFkB signaling, the activation of CNR1 was observed to decrease NFkB activity, whereas CNR1 inhibition increased NFkB signaling, with no discernible impact on neuronal cell death levels. Interestingly, this research revealed that retinal microglia, immune cells in the retina, appear to be largely unaffected by alterations in eCB signaling in both chick and mouse retinas.

These results underscore the influence of the eCB system on MG reactivity and the formation of proliferating MGPCs in the retina, shedding light on the potential implications for retinal health and therapeutic strategies, especially regarding glial responses to injuries.

### 3.5. TRP Channels

Transient receptor potential (TRP) channels constitute a superfamily of cation-permeable ionotropic receptors initially identified in the visual system of spontaneous mutants of Drosophila melanogaster. The electroretinogram (ERG) of these flies revealed a loss in the sustained depolarizing response of photoreceptors to light stimuli, contrasting with the sustained responses observed in normal flies [147]. This discovery led to the identification of a receptor named TRPC1 (canonical), encouraging the further exploration and characterization of other members within this superfamily, proving pivotal in various physiological contexts [148]. Diverse in their structure and activation mechanisms, TRP channels represent the second-largest class of ionotropic receptors described [149]. Based on their primary structural similarities, these channels have been classified into seven subfamilies: TRPA (ankyrin), TRPC, TRPM (melastatin), TRPML (mucolipin), TRPN (no mechanoreceptor C potential), TRPP (polycystin), and TRPV (vanilloid). These channels exhibit sensitivity to various stimuli, including temperature, pH, osmolarity, inflammation, membrane stretch, inorganic ions (e.g., Ca^2+^ and Mg^2+^), phosphorylation, lipids (e.g., AEA [150]), and their metabolites (e.g., arachidonic acid and epoxyeicosatrienoic acid) [151]. This diversity equips cells to detect subtle variations in both external and internal environments [152].

TRP channels have been identified in the retinas of various animals, playing crucial roles in the lower visual coding process [153]. mRNA for members of all subfamilies has been detected in the mouse retina [154]. However, the precise cell-specific localization of TRPs poses a challenge due to the limited availability of specific antibodies [154,155,156,157,158].

TRPC_1–6_ channels, excluding TRPC_2_, have been identified in the retina, with some associated with specific functions. TRPC_1′_s expression is noted in the rods, plexiform layers, INL, and vascular cells, influencing phototransduction, angiogenesis, and synaptic activity [159,160]. TRPC_3_ is present in the vascular endothelium, while TRPC_4_ is found in the Müller glia, potentially impacting angiogenesis and synaptic activity [159,161]. TRPC_5_ exhibits developmental expression in amacrine cells and Müller glia, later localizing in INL cells, such as bipolar cells, horizontal cells, amacrine cells, displaced RGCs, and Müller glia, and in both plexiform layers. TRPC_5_ influences the release of GABA by amacrine cells in the control of RGC axon length and perhaps angiogenesis [160,162]. TRPC_6_ is found in the Müller glia, RGCs, and vascular endothelium, participating in neuroprotection, angiogenesis regulation, and potentially myogenic vasoconstriction [153,159,160,161,162]. TRPM_1–3 and 7_ channels are also present in the retina, with TRPM_1_ being the most extensively studied. TRPM_1_ is detected in the rod and cone ON bipolar cells, contributing to several functions, including the depolarization of ON bipolar cells in response to light, regulation of RGC activity, development of rod bipolar cells, and establishment of synaptic connections with an amacrine cell subtype [163,164,165,166,167]. TRPM_2_ is detected in the RPE and possibly in microglia, playing a role in neuronal survival and potentially responding to oxidative stress [168,169]. TRPM_3_ is found in the RGCs and Müller glia, regulating the spontaneous activity of developing circuits [170]. TRPM_7_ is identified in the vascular smooth muscle [153,171].

TRPA_1_ is in the Müller glia, horizontal cells, amacrine cells, and RGCs, influencing redox balance and mediating neuronal damage [172]. TRPP_1_ is found in the vascular smooth muscle cells, hypothesized to play a role in myogenic vasoconstriction [171]. The location and function of the TRPML subfamily remain uncertain [153].

Except for TRPV_3_, all members of the TRPV subfamily (1–6) are present in the retina, with TRPV_1_ and TRPV_4_ exhibiting the most substantial evidence regarding their localization, function, and potential for neuroprotection [153]. TRPV_1_ is a channel primarily permeable to Ca^2+^ and Na^+^ cations and is responsive to certain vanilloids found in peppers, such as capsaicin and piperine. This channel serves as an information conduit for cells, participating in the sensory transduction of pain, touch, light, temperature (42 °C [173]), osmolarity, pheromones, acidity (pH 6.5 [174]), inflammation, and taste [175,176]. Additionally, endogenous molecules, like the endocannabinoids anandamide (AEA) and N-arachidonoyl dopamine (NADA), as well as exogenous molecules, like the phytocannabinoid cannabidiol (CBD) [177], and vanilloids, like capsaicin and piperine, also modulate TRPV_1_.

TRPV_1_ is distributed in photoreceptors, horizontal cells, bipolar cells, amacrine cells, microglia, some RGCs, the vascular endothelium, and the vascular smooth muscle. This receptor has been implicated in various functions, including the modulation of synaptic transmission, regulation of RGC function and survival, release of endocannabinoids, and control of angiogenesis. Additionally, it might be involved in the lateral inhibition and purinergic control of basal vascular tone [153,156,178,179,180,181]. TRPV_1′_s presence in the OPL [182] and photoreceptors [183] has been associated with postsynaptic transmission to bipolar and horizontal cells. This is intriguingly accompanied by the paradoxical absence of changes in the a (outer retina) and b (inner retina) waves of the photopic and scotopic electroretinogram in TRPV_1_-knockout animals [184]. Colocalization of TRPV_1_ in the IPL with synaptophysin suggests a role in presynaptic potential toward the GCL [181]. Moreover, the diffuse localization of TRPV_1_ in the OPL also implies its presence in Müller glial processes and/or resident microglia [183,185].

Animal models simulating RGC degeneration, such as those for glaucoma, hold the potential to elucidate the role of TRPV_1_ and eventually offer insights for the development of neuroprotective strategies targeting TRPV_1_. In glaucoma models induced by elevated intraocular pressures, TRPV_1′_s expression in the RGCs increases. Conversely, TRPV_1_ antagonism using iodoresiniferatoxin enhances RGC density and diminishes apoptosis induced by a high hydrostatic pressure [183], indicating a promising avenue for neuroprotection. It is intriguing to observe that TRPV_1_ undergoes diverse modulation across different cell types and species. Its modulation by exogenous agents, like capsaicin and CBD, provides valuable insights into the pharmacology governing the effects of this channel. CBD, by displacing capsaicin from the TRPV_1_ receptor and acting as its agonist, elevates intracellular Ca^2+^ levels [186]. This interaction, linked to the desensitization and internalization of TRPV_1_ [187], positions CBD as a potential pharmacotherapeutic treatment in conditions where TRPV_1′_s inhibition is crucial, such as pain, epilepsy [188], and, potentially, glaucoma. TRPV_2_ is present in bipolar cells, amacrine cells, RGCs, vascular smooth muscle cells, and in some somatostatin- and P2X7-positive cells. Its function is associated with the regulation of vascular tone and the permeability of the blood–retinal barrier [189,190].

### 3.6. Adenosine

Adenosine is an important neuromodulator in the CNS [191,192] and regulates adenylyl cyclase activity through distinct G protein-coupled receptors named A_1_, A_2a_, A_2b_, and A_3_, which are present in the retinas of several species [193,194,195,196,197]. A_1 _ receptors are expressed in the early development of the chicken retina, modulating dopamine-dependent cyclic AMP accumulation [110,198], while A_2_ receptors appear in the late stages of retinal development, promoting direct adenylyl cyclase activation [199]. Adenosine and adenosine transporters and receptors are also expressed in mixed neuronal–glial cultures of developing chicken retinal cells [200,201], and it was demonstrated that A_1_ receptor expression is dependent on cell aggregation and cyclic AMP accumulation induced by the activation of A_2a_ receptors [202]. In purified retinal neuronal cultures obtained from E8 embryos, long-term activation of A_2a_ receptors regulates the survival of neurons as well as photoreceptors [203], and protects neurons from glutamate excitotoxicity [204]. However, in cultures from E6 embryos, adenosine promotes cell death when added in the first day of culture, and this effect depends on A_2a_ receptors modulating CREB inhibition through a PKC pathway. On the other hand, the survival effect in E8 cultures is mediated through a cyclic AMP/PKA pathway and CREB activation, thus demonstrating a shift in the signaling pathways modulated by A_2a_ receptors during chick retina development [205]. Uptake and release mechanisms for adenosine were also described in chick retinal cultures [201,206], and a calcium-dependent release of purines was described in these cultures [206] when submitted to depolarization or stimulated with glutamate [207]. Interestingly, the release of purines was also found to be mediated by transporters in a calcium/CAMK II-dependent way [207]. Our recent data show the presence of adenosine A_3_ receptors modulating the release of ascorbate in cultures of chick retinal cells [196].

### 3.7. Neuropeptides: PACAP

Pituitary adenylyl-activating polypeptide (PACAP) is a neuropeptide that contains 27 or 38 amino acids and belongs to the same family as the vasoactive intestinal peptide (VIP), with which it shows high homology. This leads to common receptors that are activated by these peptides: PAC_1_, VPAC_1_, and VPAC_2_, and these are coupled to one or more signaling pathways depending on the isoform [208,209,210]. Earlier evidence of potential roles for PACAP signaling in the retina were proposed by Onali and Lianas [211], who showed that PACAP efficiently induced adenylyl cyclase activation in the retinas of various species. After that, many research groups have described critical roles for PACAP signaling in retinal development as well as in the mature retina, mostly with neuroprotective roles [212,213]. When studying the potential effects of PACAP in retinal development, we showed that it does induce the cell cycle exit of late retinal progenitors from rats through the downregulation of cyclin D_1_ [214], which correlated with the transient induction of Klf4 [215]. PACAP also contributed in the developing avian retina for the acquisition of the dopaminergic phenotype, defined by the expression of tyrosine hydroxylase [112], and interestingly, although the response to PACAP is less potent throughout development when cAMP accumulation is measured, this desensitization may be reversed through the use of a PACAP antagonist (PACAP6–38), leading to a two-fold increase in the number of TH-positive cells [216].

PACAP has also been shown to have neuroprotective and regenerative properties [212,213]. Protective effects of PACAP were described in various developmental stages, cell types, and disease models. In the neonatal retina, we showed a protective effect in both postmitotic undifferentiated cells and developing photoreceptors. In postmitotic precursors, the effect was dependent on cAMP/PKA signaling, and we detected that CREB was activated as early as 5 min after treatment [217,218]. Denes and colleagues [218] also showed that PACAP contributed to the generation of horizontal cells in the postnatal rat retinas through the induction of cell proliferation.

In disease models, the evidence of neuroprotective effects is abundant. In the intraocular hypertension ischemia-reperfusion model, one experimental model for glaucoma, intravitreal injection of PACAP protected retinal ganglion cells in the fM and pM ranges with bell-like curves. This effect was shown to be dependent on the cAMP/PKA and MAPK pathways [219]. In an ischemia model of bilateral common carotid artery occlusion, Danyadi et al. suggested functional recovery based on electroretinographic measurements [220]. In a model of oxygen-induced retinopathy (OIR) used to reproduce the retinopathy of prematurity (ROP), a protective effect of PACAP applied intravitreally in the extent of an avascular area was shown [221]. When the same model of retinopathy was applied to wild-type (WT) and PACAP KO mice, the authors showed differences in the retinal vasculature, with an enhanced avascular area and an impact on ERG [222], reinforcing that the absence of PACAP increases the vulnerability to stressors. Patko and coworkers [223] also showed the effects of PACAP applied in eye drops on the preservation of the retinal vasculature in a glaucoma model with the increase in intraocular pressure induced by microbead injections. In this study, they also demonstrated that PACAP blocked the change in the thickness of the retinal nerve fiber layer (RNFL) and total thickness of the retina [223]. PACAP also showed protective effects on UV-A-induced lesions that led to the severe degeneration of photoreceptors and impacted the inner nuclear layers and plexiform layers [224]. The evidence has also accumulated on the protective effect of PACAP in neurodegenerative diseases of metabolic origin, in particular diabetic retinopathy, as reviewed by Gabriel and colleagues [225].

Interestingly, Wang et al. used an exosome-mediated strategy for PACAP delivery in a model of traumatic optic neuropathy and showed a protective effect for RGCs, with an increase in the RNFL thickness and regeneration of axons as well as enhanced optic nerve function [226]. Recently, Van and coworkers tested whether PAC1 receptors are critical for protecting retinal neurons in a cell-autonomous manner, using adeno-associated virus (AAV2) to deliver Cre recombinase to the retina of mice harboring floxed PAC1 alleles.

Mice were challenged with a chronic form of experimental autoimmune encephalomyelitis (EAE), which recapitulates major features of multiple sclerosis (MS) and associated optic neuritis. The deletion of PAC1 under control conditions resulted in a deficit of retinal ganglion cells (RGCs) and dendrites, which unexpectedly suggests a homeostatic role of PAC1. In addition, the absence of PAC1 resulted in an increased EAE-induced loss of a subpopulation of RGCs, which had been previously described as more vulnerable in glaucoma models. Damage to axons and increased recruitment of microglia/macrophages to the optic nerve was also described [227].

### 3.8. Nitric Oxide

Nitric oxide (NO) is a gaseous signal that serves as a key regulator of various physiological processes within the retina, including transmission, vascular regulation, and immune responses [228,229]. Its production occurs through the enzyme nitric oxide synthase (NOS), which catalyzes the reaction of L-arginine, NADPH, and oxygen to form NO, citrulline, NADP +, and H_2_O [230]. In the retina, the presence of L-arginine transport systems has been described and linked to NO production since the early developmental stages, as demonstrated in chick retinal cultures [231]. Among several cell types, Müller cells uptake and deliver L-arginine to neuronal NO synthesis demand in the retina [36,231], and astroglia in the cortex [232,233,234]. There are three isoforms of NOS, two of which are constitutive and calcium/calmodulin-dependent: the neuronal (nNOS or NOS-1) [235] and the endothelial isoforms (eNOS or NOS-3) [236]. Alternatively, the constitutive isoform binds to the calmodulin- and calcium-independent inducible isoform (iNOS or NOS-2) [237]. In addition to calmodulin, four more cofactors are required for enzymatic catalysis—flavin adenine dinucleotide (FAD), flavin mononucleotide (FMN), heme, and tetrahydrobiopterin (BH4) [230].

The expression of nNOS is primarily localized in neurons, including those within the retina. Its widespread distribution and high activity would immediately provide a basis for concluding that NO should be involved in several functions within the CNS [238,239]. Pioneering work established a clear linkage between NMDA-type glutamate receptors, NO, and cGMP production in the CNS [240,241]. Functionally, NO in neurons may require a physical coupling between nNOS and NMDA receptors to compartmentalize the influx of Ca^2+^ from the channel pore to calmodulin [242]. nNOS possesses a PDZ domain, which interacts with proteins such as PSD-95 (postsynaptic density protein-95), a scaffold protein located in the postsynaptic region of neuronal cells [243].

In the retina, nNOS is localized in specific retinal cells [244,245,246,247]. This isoform was predominantly found in the puncta in the IPL, amacrine cells, and the GCL. For a detailed review, see [248]. Three main types of nNOS-positive amacrine cells have been identified, one of which is referred to as displaced amacrine (adjacent to the ganglion cell layer). All amacrine NOS-positive cells are GABAergic cells and express the GABA-synthesizing enzymes GAD-65 and GAD-67 [249,250]. These cells receive synaptic inputs from cone bipolar cells and various other amacrine cells. They also form synapses with ganglion cells, as well as with bipolar cells [248].

The distribution of nNOS plays a pivotal role in neurotransmission, synaptic modulation, and other neuronal functions within the retina. NO significantly influences neurotransmission and the modulation of signal transmission between retinal cells. It also impacts the release of neurotransmitters and synaptic plasticity [230,251]. This contribution is instrumental in the regulation of visual signal processing and adaptation to changing light conditions. For example, it has been demonstrated that light stimulation can provoke depolarizing inward currents in amacrine cells with a transient increase in intracellular calcium levels mediated by voltage-dependent channels, which would trigger the activation of nNOS [252]. The fluorescence technique, using 4-amino-5-methylamino-2′,7′-difluorofluorescein diacetate (DAF-FM), has been applied to visualize NO synthesis in the retina. This technique is useful for correlating the expression of the NOS enzyme with NO production. However, it is crucial to bear in mind that DAF-NO adducts may reflect the diffusibility of NO, given that the DAF probe can permeate various cell types and primarily serve as a target for NO trapping, rather than specifically identifying the cell type responsible for NO synthesis. In any case, most authors agree that there is a good correlation between the location of nNOS and the possible radial distance for the action of NO detected by DAF-NO adducts, which becomes a useful tool for understanding the physiology of NO retina [253,254].

Hence, physiologically, NO synthesis in the retina is regulated by light exposure and the extent of visual adaptation [255,256,257]. There is evidence that NO production in cone cells increases their responses to light during adaptation [258]. NO also appears to reduce the coupling of gap junctions between horizontal cells or even decrease the conductance of gap junctions between bipolar cells and amacrine cells of the AII subtype [259,260].

Moreover, soluble guanylate cyclase (sGC), the canonical receptor for NO (in its free radical form), exhibits high expression levels in the inner retina, but its presence in the outer retina is subject to controversy. While some authors found a very limited expression [261], others identified its presence in the outer nuclear layers and in both plexiform layers [253]. Strong immunostaining was observed in specific subgroups of bipolar and amacrine cells, with relatively weaker staining in rod bipolar cells in specific ON cone bipolar cells and, to a lesser extent, in OFF and rod bipolar cells, as well as certain ganglion cells [261]. NO donors were able in enhance cGMP, detected through cGMP immunocytochemistry visualization in the IPL and OPL and select amacrine cells, bipolar cells, and somata in the GCL [253]. Photoreceptors, horizontal cells, and Müller cells appear to not show immunoreactivity for sGC [261].

Canonical NO signaling has been shown to modulate a variety of channels and receptors, including Ca^2+^ channels [262], GABA_A_ receptors [263] and AMPA/kainate receptors [264,265].

NO can also regulate the release of several neuromodulators, as well as activity of transcription factors in the chick retina. For example, it was shown that NMDA receptor stimulation could stimulate glutamate, GABA, and glutamine release in the retina, through a mechanism entirely dependent on NO [266,267]. In the adult turtle retina, NO can also stimulate GABA release through cGMP-dependent mechanisms, which involves the reversal of GABA transporters (GAT) in horizontal cells. This process is dependent on calcium ions in the inner plexiform layer [268].

It has been demonstrated in different animal models (bovine, rabbit, and carp) that NO inhibits depolarization-stimulated dopamine release in retinal cells. Furthermore, this NO modulation of dopamine release may represent a sophisticated and high-level function in the process of light transduction within the retina, as dopamine is a recognized neurotransmitter associated with light adaptation [257].

The presence of the sodium-dependent ascorbic acid transporter (SVCT-2) has been demonstrated in the INL of rat retinas [269], as well as in cultured chick retinal cells and post-hatched chick retinas [270,271]. NO donors (SNAP and Noc-5), as well as L-arginine, stimulate ascorbic acid uptake in cultured retinas through the canonical pathway [88]. Interestingly, it was observed that this stimulation occurred through an increase in the Vmax for ascorbic acid uptake, suggesting that NO can modulate the levels of active SVCT-2 transporters on the membrane in vitro and ex vivo. This hypothesis gained strength, as it was detected through qRT-PCR, Western blotting, and immunocytochemistry analyses that NO increased SVCT-2 transcription and expression through its classical sGC/cGMP/PKG pathway [88]. This effect appears to occur through NF-κB activation since its inhibitors (PDTC and sulfasalazine) completely blocked NO- or L-Arg-induced SVCT-2 expression and ascorbic acid uptake [270].

It has also been described that NO can activate the phosphorylation of the transcription factor CREB through glutamatergic signaling. Both AMPA [272] and NMDA [37] ionotropic receptors have been implicated in this effect, which has been described as occurring via the canonical PKG-dependent pathway. Interestingly, it has been demonstrated that NO can also mediate CREB phosphorylation in Müller glia through a mechanism involving PKG and ERK-II in an evident neuron–glia crosstalk [272]. In addition, it has also been demonstrated that NO is involved in extensive cell death during the early stages of retinal development (E6), while during the subsequent stages (E8), NO significantly reduces apoptosis. In this study, NO significantly decreased nuclear phospho-CREB staining in E6, while robustly enhancing CREB phosphorylation in the nuclei of E8 neurons. The ability of NO to differentially regulate CREB during retinal development depended on the capacity of PKGII to decrease (E6) or increase (E8) nuclear AKT activation. These data demonstrate that NO/PKGII-mediated signaling may function to control the viability of neuronal cells during early retinal development through AKT/CREB activity [251]. Moreover, despite the well-known neurotoxic actions of NO synthesis, this messenger is clearly associated with neuroprotective effects in the retina [230].

Finally, even though NO signaling primarily occurs through nNOS, the retina also expresses the eNOS and iNOS isoforms. As is well-known, NO is a potent vasodilator and, in the retina, this property is vital for regulating blood flow to meet the metabolic demands of retinal cells. When there is an increased need for oxygen and nutrients, such as during increased neuronal activity, NO is released to dilate blood vessels, ensuring an adequate supply of resources to the retinal tissue. Conversely, a reduced production or availability of NO can lead to impaired blood flow regulation and potential retinal ischemia [228]. iNOS is typically not present at baseline in healthy retinal tissue but can be induced in response to inflammatory and immune stimuli. Its expression is induced by various immune and inflammatory signals, and its activity leads to the production of NO. In the retina, iNOS-derived NO is involved in immune responses and can modulate the inflammatory environment during retinal diseases or injuries. NO can have both protective and harmful effects in the retina, depending on the context. It can contribute to the regulation of immune responses during retinal pathologies, such as diabetic retinopathy, uveitis, or glaucoma [229].

In summary, NOS enzymes, including eNOS, nNOS, and iNOS, are responsible for synthesizing nitric oxide in the retina as well as its associated tissues. Each isoform has a specific cellular distribution and function, contributing to various physiological processes in visual function, such as neurotransmission, vascular regulation, and immune responses. The balanced activity of these NOS enzymes is essential for maintaining retinal function and responding to changing conditions and challenges [229].

## 4. Gliotransmitters

In the retina, ATP can be released through both vesicular and channel-mediated mechanisms. While vesicular storage and the release of nucleotides is mediated by the vesicular nucleotide transporter protein (VNUT) that is expressed in photoreceptors, bipolar and amacrine cells, Müller glia, and astrocytes in the mouse retina [273], nucleotides can be released through several channels, such as pannexin hemichannels from ganglion cells [274]. Several stimuli, including glutamate, tonicity changes, ischemia, growth factors, or purines, induce channel-mediated ATP release from RPE cells [275]. Moreover, either channel or vesicular nucleotide release from Müller glia can be triggered by mechanical/osmotic or neurochemical stimuli, such as glutamate or nucleotides themselves [276,277,278,279].

### 4.1. Nucleotide Receptors

The retina expresses several G protein coupled P2Y receptors that are mainly coupled to calcium mobilization. While the P2Y1 receptor is the main P2Y receptor in this tissue, P2Y2, P2Y4, and P2Y6 receptors were also detected [280]. Direct evidence for the P2Y11, P2Y12, and P2Y13 receptors is still missing. However, the expression of mRNA for P2Y12 receptors in the postnatal rat retina [281], as well as the blockade of glial proliferation through a P2Y13 specific antagonist [282], was obtained.

Many P2X receptors that are ion channels are also expressed consistently in the retina. P2X1–7, except P2X6, are well expressed, with P2X7 being the best-characterized subtype in the retinas of several species.

### 4.2. Nucleotides and Retinal Cell Proliferation

A major effect of nucleotides in the developing retina is the stimulation of progenitor proliferation. The activation of P2Y2/4 receptors by ATP or UTP induces the proliferation of progenitors that will generate photoreceptors, amacrine, ganglion, and horizontal cells [283,284,285]. The activation of ADP-sensitive receptors induces the proliferation of late-developing glial/bipolar progenitors [286,287] by stimulating their entry during the S phase of the mitotic cycle [288].

The nucleotide-dependent proliferation of retinal progenitors is associated with the formation of inositol phosphates [289], Ca^2+^ mobilization from intracellular stores, and its capacitive entry that occurs as early as embryonic day 3 in the chick embryo retina [283,290,291]. These responses decrease as the progenitors exit the cell cycle and begin to differentiate [292], responses that, similar to the ATP-induced increase in [^3^H]-thymidine incorporation, are decreased by conditioned medium obtained from postmitotic retinal cells in culture [287].

ADP-mediated increase in cell proliferation is inhibited by MEK inhibitors in the developing chick retina [286,289] and ADP activates the ERK pathway over the neuroblastic layer, where BrdU-labeled glia progenitors are located [289]. PI3K/Akt is another signaling pathway associated with the nucleotide-induced proliferation of retinal progenitors [293] and Müller cells from the adult retina [294,295]. In retinal cell cultures, ADP or ATP induces the phosphorylation of Akt, which increases cyclin D1 involved in the progression of cells through the G1 phase of the cell cycle [293]. Phosphorylated Akt is also observed in retinal progenitors during mitosis and is required for expression of CDK1 that controls the transition of progenitors from the G2 phase to mitosis [296].

ADP phosphorylates the cyclic nucleotide-responsive element binding protein (CREB) through an ERK-dependent mechanism that is also required for the proliferation of retinal glial progenitors in culture [282].

The nucleotide receptor subtype(s) involved in the proliferation of glial progenitors is still poorly defined [297]. Knocking down P2Y1 receptor expression decreases eye formation in frog tadpoles, and more than 80% of glial progenitors of the newborn mouse retina express P2Y1 receptors [288]. Injecting the P2Y1 receptor antagonist MRS2179 in the eyes of newborn rats decreases the number of BrdU-positive progenitors [281]. However, eye formation and retinal function were shown not to be affected in P2Y1-knockout mice [298], suggesting that other receptor subtypes may operate in the absence of the P2Y1 receptor in the developing retina. Either P2Y1 or P2Y13 receptor antagonists prevent the ADP-induced proliferation of retinal glial progenitors in culture, and the stimulation of only the P2Y1 receptor does not induce their proliferation [282], suggesting that both receptors participate in the proliferative response of chick retinal glial progenitors in culture.

In the newborn rat retina, the blockade of P2Y12 receptors induces an increase in cyclin D1 and a decrease in p57 protein. Since P2Y12′s inhibition does not affect the S phase of the cell cycle and induces the death of cyclin D1-positive cells, activating these receptors seems to be required for the exit of late-developing retinal progenitors from the cell cycle [281].

### 4.3. Nucleotides and Retinal Cell Migration

The damaged mammalian retina has a low capacity to regenerate, and de-differentiated glia contribute to the formation of glial scars. In rabbits, after retinal detachment, Müller cells migrate to the outer retina and undergo mitosis, and some cells grow beyond the OLM, forming glial scars in the subretinal space [299]. ATP may contribute to the formation of glial scars by regulating both the proliferation and migration of Müller cells [300]. Accordingly, the activation of UTP-sensitive P2Y2/4 receptors induces the growth of glial cells through a mechanism involving the PI3K, SRC, and FAK signaling pathways in mechanically scratched retinal cultures [301]. When purified retinal glial cultures are used, both cell adhesion and migration are decreased by P2 receptor antagonists [301].

### 4.4. Nucleotides and the Induction of Cell Death in the Retina

The activation of cytotoxic mechanisms by nucleotides in the developing retina was demonstrated in newborn rats and in developing avian retinal cells in culture [302,303]. The application of ATP to isolated rat retinas induces the death of cholinergic amacrine cells that express P2X7 receptors. In developing avian retinal cells in culture, P2X7 receptor-induced death of neuroblasts is dependent on the presence of glial cells and can be blocked by glutamate receptor antagonists.

Nucleotide-induced cytotoxic mechanisms were also demonstrated in mature RGCs and photoreceptors. P2X7 receptor-induced death of rat retinal ganglion cells in culture were clearly demonstrated by the authors of [304]. The sustained stimulation of these cells with the P2X7 agonist Bz-ATP provokes large increases in intracellular calcium, followed by their death. Ganglion cell death induced by nucleotides is blocked by P2X7 receptor antagonists and is also observed in the retina in vivo [305].

P2X7 receptors were implicated in the death of retinal neurons promoted by several kinds of injury. Hypoxia induces a significant increase in the death of retinal neurons in culture that can be prevented by the P2X7 receptor antagonists BBG and oxidized ATP [306]. High-pressure transients applied to rat retinas or oxygen/glucose deprivation in human retinas induce significant damage to retinal ganglion cells that is prevented by apyrase and P2X7 receptor antagonists [302,307]. Accordingly, an increase in intraocular pressure or activation of Müller cells activates microglia after ATP release and activation of the P2X7 receptor in these cells [308,309]. In rats, optic nerve crush (ONC) causes retinal ganglion cell death that is significantly attenuated when P2X7 receptor antagonists are applied during 7 days after the injury [310]. In this species, intravitreal injection of an agonist of metabotropic glutamate receptors induces Müller cell gliosis with increased ATP released from these activated cells and increased death of ganglion cells that is partially blocked by the application of the P2X7 receptor antagonist BBG, indicating that reactivation of retinal glial cells can induce the death of ganglion cells through the release of excessive ATP and activation of P2X7 receptors [311]. Interestingly, in this model, glia activation induces the upregulation of P2X7 receptor in ganglion cells through a mechanism dependent on ATP released from the activated glia, indicating that gliosis may potentiate the deleterious effect of ATP by upregulating P2X7 receptor expression in ganglion cells [311] An upregulation of P2X7 receptor expression in these cells is also observed in rat retinas from eyes submitted to an elevated intraocular pressure (IOP) [306] and during the early stages of development of the retinas of *rds* mice, a murine model of *retinitinis pigmentosa* disease [312].

The death of retinal photoreceptors induced by P2X7 receptor activation was also demonstrated. Intravitreal injection of ATP causes consistent apoptosis of photoreceptors in the rat retina, an effect that is significantly reduced by P2X7 receptor antagonists [313,314]. ATP released in the subretinal space after retinal detachment promotes the pyroptosis of microglia through P2X7 receptor activation, leading to photoreceptor death [315]. A P2X7 antagonist also slows photoreceptor degeneration in the retina of a *rd1* mouse model of *retinitis pigmentosa* [313]. In retinas from humans with age-related macular degeneration (AMD), photoreceptor cell apoptosis also occurs via P2X7 receptor activation [314].

### 4.5. P2X7 Glial Receptors and Retinal Development

Activation of the purinergic P2X7 ionotropic receptor increases calcium influx in most of the glia cells, which are substantially located in Müller glia, astrocytes, microglia, and oligodendrocytes [141,142,143]. In the avian retina, progenitor emergence around the first embryonic week is modulated by cannabinoid receptor activation (by the CB_1_/CB_2_ agonist WIN 5212-2 (WIN) [132]. Indeed, progenitors’ proliferation decreased, as assayed through [(3)H]-thymidine incorporation, when cultures were incubated with 0.5–1.0 μM WIN. In addition, the same effect was shown in the presence of URB602 and URB597, inhibitors of monoacylglycerol lipase (MAGL) and fatty acid amide hydrolase (FAAH), respectively [132,144]. In our hands, retinal cells in culture respond selectively to KCl and/or AMPA (neurons) or ATP (glia), while progenitor cells were activated by muscimol or GABA [132,145]. 

## 5. Antioxidants

### 5.1. Glutathione

Glutathione (GSH) is a tripeptide with essential redox duties in the CNS that is found at a higher concentration in glial cells [316], especially in the retinal Müller glia, compared to the neuronal compartment, which allocates ascorbate as the main antioxidant controlling biochemical processes such as protein folding, maintenance of the redox state by disulfide exchanges, and gene expression regulation [317]. It is suggestive that neurodegenerative diseases may lower the GSH/GSSG ratio, altering the levels of these peptides, and misexpress certain enzymes associated with the biosynthesis of GSH [317,318]. A low GSH/GSSG ratio leads to mitochondrial dysfunction. Nevertheless, the relationship between GSH and the glutamatergic system in the pathogenesis of nervous system diseases varies from synergism to antagonism [319]. Increasing GSH/GSSG levels systemically is obtained with the administration of N-acetylcysteine (NAC). It is important to highlight that in the chicken embryo retina, GSH induces calcium influx in cultured Müller glia but not in neurons [320,321].

GSH has been investigated for its potential roles as both an antioxidant and a signaling molecule. A study using embryonic avian retinal cells, including mixed retinal cells and purified Müller glia cells in culture, investigated the effects of GSH on calcium shifts in these cells. As shown, GSH induces calcium shifts exclusively in glial cells, which were later identified as 2M6-positive cells, while neurons responded to KCl [321]. In addition, the P2X7 receptor is involved in the effects of GSH on Müller glia. Intriguingly, GSH’s oxidized form, GSSG, fails to induce calcium mobilization in glial cells, underscoring the specific importance of GSH’s antioxidant and structural properties in elevating cytoplasmic calcium levels. Additionally, a short GSH pulse was found to protect Müller glia from oxidative damage caused by hydrogen peroxide (H_2_O_2_).

GSH was also shown to induce GABA release from various retinal cell cultures, including Müller cells, which can be inhibited by the P2X7 blocker BBG or in the absence of sodium [142]. Moreover, GSH induces propidium iodide uptake in Müller cells in culture, and this effect is mediated by the P2X7 receptor. Overall, this study suggests that GSH, in addition to its well-established antioxidant role, functions as a signaling molecule, particularly in Müller glia, regulating calcium shifts and GABA release.

The signaling properties attributed to GSH may be further corroborated by evidence showing high concentrations of this molecule in the retinas of chicks and other model animals [316]. Pow and Crook showed that rabbit Müller cells were strongly immunoreactive for GSH, while neurons presented low or undetectable levels of this molecule [322]. Although glial GSH was shown to be relevant for neuronal protection during stress [323], there is evidence to support the idea that the elevated GSH concentrations found in the retina are not directed to enhance cell survival [324]. In fact, Castagné and Clarke showed that the inhibition of GSH synthesis by L-buthionine-[S,R]-sulfoximine can diminish retinal cell death [325].

### 5.2. Vitamin C

Vitamin C, made up of its oxidizing and reducing components ascorbate (AA) and dehydroascorbate (DHA), respectively, is essential for multiple physiological functions. Many mammals are capable of synthesizing vitamin C from glucose; however, humans do not have the last enzyme responsible for its biosynthesis [326]. As a result, vitamin C must be ingested through food and supplements. Once absorbed, vitamin C will be distributed to tissues through its transporters, which are of two types: sodium-dependent vitamin C transporters (SVCTs), which transport AA, and glucose transporters, which transport DHA [327]. High concentrations of vitamin C are found in the brain, mainly in neuronal cells [328]. Among the physiological processes, vitamin C acts as an enzymatic cofactor in the conversion of dopamine into noradrenaline [329], a reducing agent and scavenger of oxygen- and nitrogen-free radicals generated during cellular metabolism, increases synaptic activity [330], participates in the formation of the myelin sheath for Schwann cells [331], and acts as a neuromodulator in the nervous system. As neurodegenerative diseases are associated with high levels of oxidative stress, AA has been considered an important therapeutic agent for neurodegenerative diseases. Studies have shown that the pathophysiological processes of neurodegenerative diseases and neuropsychiatric disorders are improved with nutritional interventions. Among them, the association of treatments with AA has presented a promising scenery. The anti-inflammatory, antioxidant, and anti-excitotoxic role of ascorbate is believed to be responsible for its protective actions [332,333,334,335].

## 6. Reciprocal Interactions between Retinal Transmitters

Due to the organization of the retinal tissue and the massive presence of different types of synapses, especially in the plexiform layers, it is highly expected to have an extensive interaction between these modulatory systems. In many cases, interactions are reciprocal and show distinct levels of complexity during development. Below, you can find some examples of these interactions in the chicken retina.

### 6.1. Dopamine and Adenosine

Dopamine promotes the accumulation of cAMP in developing chicken retinas since embryonic day 7 (E7), with the maximal effect observed in E8 and decreasing in subsequent days. The stimulation level in post-hatched (PH) retinas is low [336]. On the other hand, adenosine only promotes cAMP accumulation in this tissue after E13, increasing up to E17 and attaining low levels in PH, similarly to what happens with the dopamine stimulus [199]. Interestingly, adenosine can inhibit cAMP accumulation induced by dopamine since early developmental stages when direct stimulation with adenosine is no longer observed [110]. This inhibitory effect is mediated by A1 receptors, which are present during the early embryonic stages [198]. The mechanism of inhibition as well the functional and embryological consequences remain to be investigated.

### 6.2. Glutamate and Adenosine

Glutamate is a major excitatory neurotransmitter in the retina, including the chicken retina [337], where it was found to regulate the release of adenosine, GABA, and vitamin C [92,207,271]. The activation of ionotropic glutamate receptors, such as AMPA, kainite, and NMDA receptors, can promote a dose-dependent release of purines in cultures of chick retinal cells. Interestingly, this release was not only shown to be calcium-dependent but also mediated by nucleoside transport and is regulated through a calmodulin-dependent kinase type II (CAMKII) mechanism. Adenosine, but not GABA or choline uptake in the cultures, is also modulated by CAMKII, supporting the hypothesis that the enzyme directly or indirectly modulates nucleoside transport [18].

### 6.3. Glutamate and Vitamin C

Glutamate is also able to induce ascorbate (AA) release in cultures of developing chick retinal cells [271]. As stated in a previous section, ascorbate transport is mediated by the SVCT, which is expressed in the chicken retina as well as in retinal neurons in culture. The release of glutamate is dependent on the presence of sodium ions and blocked by sulfinpirazone, a SVCT inhibitor, indicating that it is mediated by the SVCT working in the opposite direction. The hypothesis is that glutamate activates ionotropic receptors, allowing for the entry of sodium ions and their accumulation in the vicinity of the SVCT, producing its functioning in the release direction [271]. Interestingly, the released AA inhibits glutamate transport through the excitatory amino acid transporter type 3 (EAAT3) present in neurons, promoting an accumulation of extracellular glutamate, activation of NMDA and AMPA receptors, and consequent activation of the signaling pathways, leading to CREB stimulation [338].

### 6.4. Glutamate and GABA

The interactions between the major amino acids glutamate and GABA were described in the CNS, including in the retina, where disturbances in the balance of these two neurotransmitters are involved in neurodegeneration and aging [339]. As stated above, the activation of ionotropic glutamate receptors or depolarization with veratridine promotes a transporter-mediated release of GABA in cultures of chicken retinal cells [92,340]. Interestingly, the transport of GABA by glial cells is regulated through a glutamatergic input, suggesting an interplay between neurons and glial cells in the retina [341]. In addition, GABA and glutamate regulate GAD expression, the main synthesizing enzyme responsible for GABA synthesis in cultured retinal cells [342], indicating a strong interaction between these neurotransmitters in the retina.

### 6.5. Dopamine and Glutamate

Dopamine regulates Src kinase activity in cultured chicken retinal cells through D1 receptors, accumulation of cyclic AMP, and activation of PKA, which phosphorylates the C-terminal domain of Src kinase (CSK) at position serine 364 [343]. Stimulation of CSK then leads to Src phosphorylation at the inhibitory domain tyrosine 527 and consequent inhibition of Src kinase activity [344]. It is well known that Src phosphorylates the N2B subunit of NMDA receptors at tyrosine 1472, thus regulating receptor function [345]. We were able to show that the activation of dopamine D1 receptors inhibits NMDA receptor function through this pathway in the retina, showing a possible important pathway linking the activation of dopamine receptors and the inhibition of glutamate NMDA receptors [343].

### 6.6. Endocannabinoid and Dopamine

Dopamine is found in amacrine retinal cells very early on during development, around embryonic day 8 [100]; on the other hand, cannabinoid receptors also emerge early on during the embryonic stages [118], which control excitability and the levels of secondary messengers, such as cAMP or calcium signaling, during development. The CB1 receptor is highly expressed from embryonic day 5 (E5) until post-hatched day 7 (PE7), decreasing its levels throughout development. While CB1 is heavily located in the GCL and inner plexiform layer (IPL), the CB2 receptor is primarily placed in the inner plexiform layer (IPL) at PE7. Cannabinoid CB1 and CB2 are found in both neurons and glial cells, but MAGL, the enzyme that degrades 2-AG, is only expressed in Müller glia [118]. TH, the regulatory enzyme that synthesizes catecholamines, is found in amacrine cells that also express both CB1 and D1 receptors. As cAMP is a signaling messenger increased by D1 activation and decreased by CB1 activation, this seems to be an important relay to regulate retinal signaling and development [118]. Indeed, neurite outgrowth has been shown to not only be modulated by cAMP in the retina but also in the entire CNS [346,347]. In conclusion, a relationship between the endocannabinoid and dopaminergic systems is found in avian retinal development that defines cAMP accumulation via D1 receptor activation and may influence embryological parameters during avian retinal differentiation [119].

### 6.7. Dopamine, Glutamate, and Vitamin C

Dopamine is also able to promote AA release in chick retinal cultures, an effect promoted by the stimulation of D1 receptors, the accumulation of cyclic AMP, and the activation of exchange protein activated by cAMP (EPAC 2) [348]. Interestingly, AA release is mediated by the SVCTs since it is sodium-dependent and blocked by sulfinpirazone. However, more recent evidence indicates that the release of AA induced by dopamine is mediated by glutamate and the activation of AMPA receptors but not NMDA receptors [349]. These results point to the existence of neuronal circuits comprising dopaminergic, glutamatergic, and AA-releasing cells in the retina.

### 6.8. Adenosine, Vitamin C, and Nitric Oxide

As described above, glutamate can release purines (including adenosine) in the retinal cultures [207], and adenosine regulates the cAMP accumulation induced by dopamine in the retina [110]. Dopamine also promotes the release of adenosine [350] as well as a glutamate-mediated vitamin C release [349]. Recent work shows that adenosine acting on A3 receptors promotes the release of AA and controls the redox balance in retinal neurons in culture [196]. Nitric oxide is also another important neuromodulator in the retina, specially linked to the activation of glutamate receptors. For example, nitric oxide regulates the SVCT in retinal cultures, increasing its expression through an NFkB-dependent mechanism [270]. Many effects of glutamate mediated by ionotropic receptors also involve nitric oxide production [351]. Indeed, some effects of vitamin C are mediated through the accumulation of glutamate and production of nitric oxide. These findings clearly indicate the existence of reciprocal interactions among different neurotransmitters and neuromodulators in the retina.

## 7. The Diseased Retina

Retinal diseases encompass a diverse range of ocular disorders that have a profound impact on visual health and patient well-being. They can range from genetic to non-genetic disorders and degenerative conditions that can lead to varying degrees of vision impairment and, in some cases, even blindness. Genetic retinal degenerations represent a significant subset of retinal diseases with an incidence of approximately 1 in 3000 individuals (https://web.sph.uth.edu/RetNet/, accessed on 20 October 2023) [352], affecting more than 2.5 million people worldwide, posing a significant burden on global eye health. As our understanding of the genetic basis of retinal diseases continues to expand, the identification of causative genes and genetic variants has been increasing, and, today, we have 341 genes with causative variants identified, presenting highly distinct disease courses and phenotypes [353]. Although an extensive genetic landscape has already been mapped, the mutations responsible for 30% to 50% of cases of inherited retinal diseases remain undisclosed [354].

Non-genetic retinal diseases constitute a diverse array of ocular disorders with a global impact, often regardless of an individual’s genetic constitution. These conditions can stem from a variety of factors, including the natural aging process, lifestyle choices, infections, and environmental influences. Age-related macular degeneration (AMD) stands out as one of the most prevalent non-genetic retinal diseases, impacting 196 million people in 2020, with a prediction to increase to 288 million people worldwide in 2040, particularly among the elderly population [355]. It leads to central vision loss through a combination of the accumulation of deposits (drusen) and, in the non-neovascular form, retinal pigment epithelium abnormalities or, in the neovascular form, abnormal blood vessel growth, both in the macula, leading to damage and the impairment of the central vision [356]. Similarly, diabetic retinopathy, intricately tied to diabetes, is another widespread non-genetic condition characterized by the deterioration of retinal blood vessels, potentially leading to severe vision loss if left untreated [357]. Retinopathy of prematurity (ROP), predominantly affecting premature infants due to excessive oxygen exposure during early medical care, is another example, known for its capacity to induce vision problems or even blindness [358]. Additionally, glaucoma, often associated with elevated intraocular pressure, is a global illustration of non-genetic ocular diseases [359]. These various non-genetic retinal diseases underscore the importance of regular eye examinations and proactive healthcare measures to prevent or manage vision impairments on a global scale.

In the face of the challenges posed by neurodegenerative conditions leading to blindness, the field of ophthalmology and vision science continues to push the boundaries of medical research. The convergence of genetics, innovative therapies, and cutting-edge technologies offers a ray of hope for those affected by these diseases. The relentless pursuit of new treatments, including gene therapies, stem cell transplantation, and precision medicine, is promising. These advancements hold the potential to not only slow the progression of vision loss but also, in some cases, to restore sight. All of these new and innovative therapies are on the cusp of revolutionary breakthroughs that may one day conquer the challenges posed by neurodegeneration and blindness, offering a brighter future for those impacted by these conditions.

### 7.1. Glaucoma

Glaucoma is an optic neuropathy characterized by an insidious onset and gradual progression that comprises a group of neurodegenerative diseases marked by structural damage to the optic nerve with axonal loss and RGC degeneration by apoptosis [360]. It is the leading cause of blindness around the globe, and the elderly are more susceptible to developing this disease. Aging populations have been increasing in both developed and developing countries, and the rise of diseases associated with aging can impact the quality of life of individuals and economic growth. Today, glaucoma is considered a major public health problem and the third main cause of long-term disabilities experienced by individuals [361]. Its prevalence varies by geographic region and demographic factors, with higher rates observed among people of African, Asian, and Hispanic descent. The number of people with glaucoma worldwide is estimated to be 80 million, increasing to a predicted 111.8 million in 2040 [362]. In Brazil, the cases rose from 900,000 in 2010 to 2.5 million in 2020 [363]. Alarmingly, up to 40% of glaucoma patients can progress to blindness. Glaucoma is highly heritable, and a true family history of glaucoma increases the risk in a first-degree relative nearly eight times compared with the general population [364]. Elevated intraocular pressure is a major risk factor in glaucoma. It is widely believed to contribute to the compression and damage of the optic nerve fibers, thereby accelerating the degeneration of RGCs. However, it is important to note that not all the individuals with a high IOP develop glaucoma, and, conversely, glaucoma can occur in those with a normal IOP, suggesting that other factors play a crucial role [365]. However, existing treatment paradigms focus on reducing the intraocular pressure, mainly through the daily use of eye drops, sometimes associated with side effects or invasive surgical interventions [366,367]. The frequent inadequacy of ocular pressure-lowering approaches, substantial rates of treatment non-adherence affecting 30–70% of patients, and resulting financial burdens indicate the need for long-lasting, neuroprotective therapies.

The increased pressure induced by glaucoma can cause ischemic events in the retina, and in the literature, there is some evidence of cannabinoids mediating or regulating the damage after these events. The blockage of CB_1_ and CB_2_ in an *ex vivo* model of ischemia can decrease the damage induced by oxygen and glucose deprivation (OGD) [140]. In an interesting way, using the same model, the blockage of TRPA_1_ (transient receptor potential ankyrin 1), a channel that can be activated by cannabinoids and their substrates, inhibits the damage induced by OGD [140]. After the role of TRPA_1_ on an acute model of glaucoma was shown, it was also demonstrated that TRPA_1_^-/-^ knockout mice show a complete blockade of retinal damage (inhibiting the decrease in retinal thickness, increase in oxidative stress, and increase in the level of caspase activity) in a model of increased intraocular pressure and 2 or 7 days of reperfusion [172].

### 7.2. Diabetic Retina

Diabetes mellitus represents a major public health problem. In 2017, the International Diabetes Federation estimated that 425 million people had diabetes, and expected the number of people affected to be approximately 629 million by 2045. Diabetic retinopathy (DR) is a microvascular complication of diabetes mellitus and is the most common cause of blindness, affecting the working age population [368,369,370]. The increasing knowledge of the pathophysiology of this disease allowed for its identification at an even earlier stage, increasing the possibility of treatment. Around 126.6 million people worldwide were affected by this condition in 2011, and this number is expected to rise to 191 million by 2030 [371], with 56.3 million of them at high risk of visual impairment, a group that includes individuals with proliferative diabetic retinopathy and diabetic macular edema [372], making this disease a major burden on the healthcare system, with a predicted increase in the number of people suffering from visual impairment [373].

Diabetic retinopathy is clinically characterized by vascular alterations, and it is divided into two stages—non-proliferative (NPDR), which represents the initial stage of DR, when microaneurysms, small hemorrhagic spots, and exudates can already be observed on fundus examination, but the patient may be asymptomatic, and proliferative (RDP), a more advanced stage characterized by neovascularization and greater visual consequences with the expansion of poorly formed vessels into the vitreous, increased risk of retinal detachment, and formation of macular edema [370].

However, many studies have demonstrated that DR is a neurovascular disease, with changes in neuronal morphology, reduction in the number of synaptic proteins, changes in neurotransmitter systems, and neuronal death in the retina [374,375,376,377,378,379]; for a review, please read [374,380]; even before vascular alterations occur and can be clinically detected [381]; for a review, see [382]. Early changes in the electrical activity, thickness of the human retina, and electrophysiological and visual perceptual changes prior to vascular changes have also been shown [75,383]; for a review, please read [382]. The RPE also showed oxidative stress [4,384], tight junction destruction [385,386,387], and apoptosis [388]. Therefore, all cell types that constitute the retina are affected by persistent hyperglycemia in diabetes and contributed to its disastrous outcome.

Several pathways have been associated with the development of DR induced by hyperglycemia/diabetes: the formation of advanced glycated end-products (AGEs), an increase in the polyol pathway, diacylglycerol/PKC, and the hexosamine pathway. However, they all have a common axis of activation: oxidative stress [389,390,391]. Therefore, antioxidative stress has become a promising strategy for the treatment of DR [391]. Although there are several preclinical studies showing promising results with antioxidants, clinical trials have been scarce with controversial outcomes (e.g., see [23] for a brief review). The nuclear factor erythroid 2-like 2 (Nrf2)/Kelch-like ECH-associated protein 1 (Keap1) pathway is a crucial pathway to fight oxidative stress (Figure 2). Nrf2 is a transcription factor that binds to a specific promoter region, the antioxidant-responsive element (ARE), stimulating the transcription of several cytoprotective genes that triggers a cellular antioxidant response, a crucial pathway to fight oxidative stress. Under physiological conditions, Keap1 binds to Nrf2, leading to the ubiquitination and degradation of Nrf2, which controls the levels of this transcription factor. With the increase in levels of reactive oxidative species (ROS), Nrf2 dissociates from Keap1 and translocates to the nucleus, stimulating the transcription of ARE-containing genes. Therefore, oxidative stress stimulates the increase in Nrf2 stability and nucleus levels, which can induce an antioxidant response. However, it has been systematically shown, in in vitro and in vivo studies, that exposure to a high level of glucose induces a decrease in Nrf2 levels, particularly in the nucleus, but also total Nrf2 in all types of retinal cells: RPE cells, mainly investigated in ARPE-19 cells [392,393,394], endothelial cells, mainly HREC cells [395], Müller cells [23,396], and ganglion cells [397]. In vivo experiments also showed that diabetes decreases Nrf2 retinal content even during the early periods of diabetes before vascular alterations occur [378,379,398,399]. As mentioned earlier, Nrf2 controls the gene transcription of several antioxidant signals, some of them crucial to glutathione (GSH) generation, such as glutathione peroxidase and the catalytic subunit (xCT) of the cystine/glutamate antiporter X_c_^−^ system (see [23] for a review). This transporter uptakes cystine, an important and limiting precursor for GSH synthesis [400]. So, a decrease in Nrf2 in high glucose conditions, in retinal cell culture, or in diabetes leads to a reduction in glutathione peroxidase, xCT, and, consequently, glutathione retinal levels, as well as in different retinal cell types (RPE, endothelial, Müller, and ganglion cells) [401].

Retinas from diabetic animals have less Nrf2 bound to the promoter of the catalytic subunit of the X_c_^−^ system (xCT) since the early stages of diabetes [379], which could explain the lower expression of xCT. Additionally, Nrf2 also controls the gene transcription of other important antioxidant enzymes, such as hemoxigenase, NQO1, and catalase, which are also decreased in retinal or other cell types in diabetic retinas or cultures exposed to high glucose in a Nrf2-dependent mode [402]. Accordingly, an increase in oxidative stress, followed by and dependent on Nrf2 reduction, is observed in the retinas of diabetic animals as well as in the cell types. Although high glucose exposure and diabetes induce oxidative stress, which activates the Nrf2 pathway, most of the studies show that the maintenance of hyperglycemia induces the reduction of Nrf2, hampering the cell capacity to fight oxidative stress and leading to cell death through ferroptosis [393] and apoptosis [401]. The apparently contradictory effect lies in a lot more complex Nrf2-regulating signaling pathways. It has been shown that Nrf2 is closely regulated by the Akt/GSK3 pathway [403,404]. Nrf2 degradation/nuclear extrusion is activated by GSK3b, which is blocked by Akt phosphorylation and the inhibition of GSK3 [403,404]. Several studies reported a reduction in Akt activation under high glucose or diabetes conditions [392,397], which can be via the classical PTEN/Akt pathway. PTEN activity is increased in hyperglycemia and diabetes, decreasing the activated Akt level [392]. In addition, it was shown that hyperglycemia increases PP2A activity, which dephosphorylates Akt and stimulates GSK3b [405]. Hyperglycemia/oxidative stress stimulates the regulation of development and DNA damage 1 (REDD1), a stress-induced protein that promotes the association of PP2A and Akt, decreasing Akt phosphorylation and activity, and inhibiting GSK3b.

In diabetes, or high glucose conditions, REDD1 levels augment and induces Nrf2 degradation through GSK3 activation [406,407], hampering the Nrf2-induced antioxidant response. REDD1 is also directly activated by oxidative stress, generating positive feedback, worsening the level of oxidative stress [407]. The Akt inhibition by REDD1 leads to a decrease in the activity of mTOR, which disinhibits 4E-BP1 that represses VEGF mRNA translation [408].

Since hyperglycemia increases the level of REDD1, a consequent increase in oxidative stress and VEGF levels is seen, contributing to angiogenesis and cell death [409].

Finally, in ARPE-19 cells, Nrf2 can also be positively regulated by SIRT-1 and AMPK [393,410]. Therefore, hyperglycemia in diabetes, or high glucose exposure in vitro, can induce, through different mechanisms, a decrease in Nrf2 levels and an impairment in the antioxidant Nrf2-stimulated response.

The inflammatory component also appears to be a central event in the progression of DR. The increase in the expression levels of adhesion molecules, such as ICAM-1, VCAM-1, and E-selectin, added to higher rates of leukocyte adhesion and leukostasis are phenomena observed in animal models of diabetes and human patients and are associated with damage to the blood–retinal barrier and the loss of endothelial cells [411,412,413,414,415]. An increased expression of chemokines, such as MCP-1, MIP-1α, and MIP-1β, and cytokines, such as TNF-α, IL-6, IL-8, and IL-1β, also appear to be involved in the pathogenesis of DR [416,417,418,419]. Glial cells in the retina, such as astrocytes, MG, and microglia, orchestrate the inflammatory reaction, producing and releasing the aforementioned factors [420,421]. Importantly, it has been shown that the increase in inflammatory cytokines, mainly TNF-α, IL-6, and IL1-β, occurs due to the Nrf2 decrease induced by high glucose or hyperglycemia in diabetic animals, and preventing Nrf2 inactivation prevents the inflammatory response [401,422].

Due to the crucial role of Nrf2 in controlling oxidative stress and, consequently, inflammation and cell death signaling pathways, like ferroptosis and apoptosis, preventing, acting directly or indirectly, the reduction of Nrf2 protects retinal cells from degeneration [393,397,401,422]. The present medical treatments for DR include glycemic control, laser therapy, glucocorticoid therapy, anti-VEGF intraocular injections, etc., but none of them cure nor stop the disease progression, and all combat the vascular alterations seen in the very advanced stage of the disease [423]. Although the available treatments are important for ameliorating the clinical deficits, new approaches, especially those that can be used earlier, are necessary to improve treatment and avoid blindness.

Drugs that inhibit important pathways for the progression of this disease have been tested for their treatment. Examples of this were tests with aminoguanidine, an inhibitor of the advanced glycation species pathway [424], to inhibit inflammatory pathways, and the systemic administration of several antioxidants [425,426,427]. Other drugs that have been tested include other antiangiogenic drugs that modulate other factors, such as PDGF, b-FGF, Ang-1, 2, and the Ang-1,2 receptor, Tie2, and drugs with anti-inflammatory effects, such as corticosteroids (which show improvements in symptoms of diabetic retinopathy but can cause increased intraocular pressure and cataracts), integrin and interleukin 6 inhibitors, alpha-lipoic acid (a mitochondrial antioxidant), lutein (a carotenoid with antioxidant action), ARA290 (a peptide derived from erythropoietin), and darapladib (a phospholipase A2-associated lipoprotein inhibitor—LpPLA-2) [370]. However, the results of these clinical studies were either inconclusive or suspended due to their side effects.

Recently, several preclinical studies have been focusing on searching for substances that inhibit the impairment in the Nrf2 pathway induced by diabetes in neural and vascular retinal cells: acteoside [428], maslinic acid [4], astragaloside IV [393], urolithin A [422], hydroxysafflor yellow A [392], carnosol [395], astaxanthin [398], and amygdalin [429], among others. However, as for the other previous approaches, it will be critical to investigate the protective ability of these agents in diabetic patients and their potential of causing deleterious side effects.

## 8. Investigation of Innovative Therapeutic Strategies

### 8.1. Neuroprotection

The Na^+^/K^+^ ATPase (NKA) enzyme, located in the plasma membrane of mammalian cells, has long been recognized for its classical function in actively pumping Na^+^ and K^+^ against their respective concentration gradients using the energy provided from the hydrolysis of ATP. This enzyme has a highly specific binding site for hormones called cardiotonic steroids like ouabain. The NKA enzyme exhibits a biphasic response to ouabain: while high (micromolar) concentrations of ouabain inhibit the NKA enzyme’s pumping function, low (nanomolar) concentrations have no immediate effect on ionic transport but can initiate several intracellular signaling cascades through protein–protein interactions (reviewed by [430]).

The functional unit of the NKA enzyme consists of a catalytic subunit, alpha, along with two regulatory subunits, beta and gamma. There are four isoforms of the alpha subunit, denoted as alpha 1 to 4 [431]. In the retina, the NKA enzyme is expressed by all cell types, exhibiting differential patterns in both adult and developmental stages [432,433]. Specifically, in the adult retina, the alpha 1 subunit is found in Müller and horizontal cells; alpha 2 is particularly present in Müller glia; and alpha 3 is detected in photoreceptors and other retinal neurons [432]. The NKA enzyme plays a crucial role in diverse retinal functions, including controlling the Na^+^ and K^+^ gradients associated with photoreceptor dark current, maintaining the resting membrane potential in RGCs, facilitating neurotransmitter uptake by Müller cells, regulating synaptic activity and light adaptation, and preserving elements of the neuro–glio–vascular unit [434,435,436,437]. The biphasic effect of ouabain also extends to retinal physiology. Inhibiting the NKA enzyme’s pumping function using high concentrations of ouabain induces extensive death of retinal neurons in animal models [437,438,439]. On the other hand, NKA-mediated signal transduction through low concentrations of ouabain can prevent RGC death [440].

In retinal pathologies, optic nerve degeneration leading to the death of RGCs is a hallmark of conditions like glaucoma and traumatic optic neuropathy, resulting in irreversible blindness and the disruption of essential physiological processes regulated by these cells [441,442,443]. For instance, in glaucoma, the affected RGC populations undergo a prolonged degenerative process, as observed in studies by [444]. For this reason, understanding the complexity of RGC death and survival mechanisms stands out as a pivotal topic worthy of discussion. Additionally, acute injuries to the optic nerve, such as axotomy or transection, are induced in animal models to investigate the processes governing the death and survival of these cells. Various in vivo and in vitro models, including those developed by the authors of [445,446], have been instrumental in studying RGC survival and optic nerve regeneration [447]. In order to prevent or delay the death of these cells, different neuroprotective mechanisms have been identified, such as the secretion of neurotrophic factors, the activation of antioxidant enzymes, the increased expression of anti-apoptotic proteins, the induction of autophagy, and proteostasis regulation [448,449,450,451]. These mechanisms can modulate cellular responses to stress, inflammation, ischemia, and apoptosis, and thus increase the survival of RGCs.

Signaling pathways regulated by the NKA enzyme in the retina are poorly understood. However, studies have indicated that ouabain promotes RGC survival after optic nerve axotomy through NKA-mediated activation of Src kinase and subsequent transactivation of the epidermal growth factor receptor, as well as the activation of the PKC delta and c-Jun N-terminal kinase signaling pathways [440,452]. In retinal cells, recent evidence indicates that ouabain prevents RGC death through stimulating autophagy [453]. In addition, ouabain also decreases oxidative stress, modulates microglial reactivity, and regulates the expression of cytokines such as brain-derived neurotrophic factor, TNF-α, and interleukin 1-β [453,454], which are critical for neuronal survival and glial functions. Thus, future studies on the physiological functions of NKA-mediated neuron–glia signaling, as well as in retinal disease models, may contribute to the development of new therapeutic strategies against retinal degeneration.

### 8.2. Gene Therapy and the Future of Vision Recovery

Gene therapy is a groundbreaking medical field that has achieved remarkable progress over the past two decades, providing newfound hope for the treatment of previously untreatable and hereditary diseases. Among the numerous applications within gene therapy, retinal gene therapy emerges as a particularly promising avenue, offering a potential solution to a broad spectrum of ocular disorders and vision impairments. The eye, with its unique characteristics, presents a compelling organ for gene therapy. Its privileged immune status makes it an ideal candidate for genetic interventions, decreasing potential immune responses to gene therapy treatments [455]. Moreover, the eye’s accessibility for medication delivery is unmatched, allowing for targeted and minimally invasive interventions. It is a critical consideration, given the complex structure and sensitivity of ocular tissues. Among these tissues, the retina is a primary candidate for gene therapy. Its visibility enables precise monitoring and evaluation, which is essential for assessing the effectiveness of gene therapies. The absence of lymphatic vessels, a direct blood network in the outer layers, and a lack of cell division post-birth make the retina an ideal canvas for achieving sustained transgene expression.

In recent years, substantial improvements have been made in identifying the genes responsible for genetic retinal diseases [355], thanks to advanced techniques like next-generation sequencing, single nucleotide polymorphism microarrays, and comparative genomic hybridization. This burgeoning genetic knowledge has laid the foundation for more precise therapeutic interventions, moving beyond symptom management towards targeting the root causes of these conditions.

The advent of gene therapies has ushered in a new era of hope for individuals afflicted by monogenic eye diseases. Ophthalmology has been at the forefront of gene therapy research, capitalizing on the eye’s unique characteristics for effective gene delivery. In 2017, the approval of voretigene neparvovec-rzyl (Luxturna) marked a significant milestone, becoming the first FDA-approved in vivo gene therapy for RPE65-associated biallelic variants. Luxturna belongs to the concept of gene replacement or augmentation, in which a functional copy of a damaged, non-functional gene is added to augment the production of functional protein, being a natural fit for inherited retinal diseases caused by loss-of-function mutations. This achievement with Luxturna has not only transformed the treatment landscape but has also ignited a flurry of research activities in the field of ocular gene therapy [456].

Subsequently, this field expanded its horizons, recognizing that many complex diseases are not only influenced by the primary disease-causing gene but also by modifier genes that can either exacerbate or mitigate the condition’s effects [457,458]. The concept of modifier gene therapy is now at the forefront of research, aiming to fine-tune the treatment of multifactorial disorders, like glaucoma, diabetic retinopathy, and macular degeneration, among others, by targeting genes that play a pivotal role in disease progression. OCU400 (Ocugen Inc.) is a modifier gene therapy to treat people with inherited retinal diseases, retinitis pigmentosa, caused by a broad range of genetic mutations, and is currently in clinical trial phase 2. The therapeutic candidate is an adeno-associated virus serotype 5 (AAV5) containing the gene for the human nuclear hormone receptor NR2E3, and as a modifier gene therapy, it expands the patient reach, treating multiple mutations with a single product instead of developing a product for every mutation, and potentially decreasing costs [459]. This evolution signifies a shift towards more precise and personalized approaches, with the goal of not only treating symptoms but also addressing the root causes of complex diseases, ultimately paving the way for more effective and tailored therapeutic interventions.

Gene therapies using CRISPR/Cas9 technology have also been assessed in clinical trials. In these gene editing therapies, the mutations in a gene are corrected or the expression of the mutated protein is reduced to alter a diseased state. In early 2020, an open-label, single ascending dose study started to enroll LCA10 patients to test the CRISPR-Cas9 gene edition to correct the IVS26 mutation (NCT03872479) by delivering highly specific small guide RNAs to the gene CEP290, along with SaCas9 under the control of a photoreceptor-specific GRK1 promoter, packaged into an AAV5 vector into the subretinal space [460]. In 2022, the developers released some results in which three out of fourteen patients showed clinically meaningful improvements in best-corrected visual acuity. These results provided a proof of concept that CRISPR-based gene editing can be safely delivered to the retina; however, the developers have made the decision to pause enrollment while looking for partners to continue the studies.

In advanced cases of retinal degeneration, in which the photoreceptors are very compromised, optogenetics comes as an innovative tool, involving the delivery of light-sensitive microbial opsins to the remaining retinal cells using gene therapy [461] With optogenetics, it is possible to treat the disease independent of the underlying gene defect. It provides new photosensitive genes, such as channel rhodopsin, halorhodopsin, and melanopsin, to the retina’s output cells, ganglion cells, or bipolar cells, adding the light-activity to these cells in their existing neural networks [462,463]. Promising results in preclinical rodent and non-human primate models led to the development of different clinical trials (NCT05417126, NCT04945772, NCT04945773, NCT02556736, and NCT03326336) less than a decade after the first attempt at visual restoration using this approach. However, optogenetics still require optimization to allow for complex visual processing and to increase the sensitivity of the photosensitive proteins that are currently in use.

The future of vision recovery through retinal gene therapy holds great promise but also presents a complex landscape of challenges. Advances in vector design are expected to prioritize reducing immunogenicity, enhancing target specificity, and improving transduction efficiency, with potential shifts towards less immunogenic vector options. Overcoming the payload size limitations of vectors may involve innovative strategies, such as non-viral techniques or dual/triple transduction methods. While the retina’s immune privilege makes it an ideal candidate for gene therapies, it introduces unique obstacles, including the identification of disease-causing genes, precise delivery, optimal administration routes, clinical feasibility, and managing immune responses. Additionally, physically delivering therapeutic products to the delicate and isolated retinal tissue remains a formidable challenge. Nevertheless, ongoing research and a multitude of creative approaches demonstrate the determination of the scientific community to unlock the full potential of retinal gene therapy, offering hope for the restoration of vision and a brighter future for individuals with retinal diseases.

### 8.3. Cell Reprogramming

Diseases that affect the retina usually lead to visual loss, which is extremely debilitating. Many research groups are now investing in the study of innovative therapeutic approaches to stop or delay disease progression, protect the affected cell populations, or even promote the regeneration of the retina for the reversal of progressive blindness. These regenerative approaches are either directed to the generation of new neurons ex vivo for transplantation [464] or to the generation of the affected neurons from endogenous cell sources, such as MG [465]. In teleost fish, MG acts as a multipotent stem cell, which, in response to damage, dedifferentiates generating MG-derived progenitors, which then give rise to all retinal cell types [466]. However, this regenerative potential is virtually absent in mammalian retinas [465,467]. Recently, Hoang at al described the differential activation of signaling pathways in response to damage in fish, chick, and mouse retinas [468]. They showed that nuclear factor I (NFI) transcription factors maintain and restore quiescence in mammalian MG, while in the zebrafish and chicks, they are essential for regeneration when the MG transit from quiescence to reactive [468].

Many research groups are investing in identifying and testing reprogramming strategies to reactivate the regenerative potential of Müller glia mainly through the modulation of the expression of specific transcription factors. Using transgenic mice to overexpress the pro-neural transcription factor Ascl1 alone and in combination with damage, Dr Reh’s group showed the generation of new neurons from the MG, which are mostly bipolar cells [469,470]. In addition, when Ascl1 was used together with a histone deacetylase inhibitor, they were able to generate bipolar cells from the MG of adult-damaged retinas, and these new neurons formed synaptic connections [471]. Recently, Todd and coworkers generated retinal ganglion-like cells (RGC-like cells) with the combination of Ascl1 and another pro-neural bHLH factor: Atoh1 [472]. When Pou4f2 and Isl1 were added to this equation, more molecular characteristics of the RGCs [473] were obtained, although no demonstration of the ability of these cells to project axons to their brain targets were shown.

On the other hand, studies using AAV vectors have also presented data on the generation of retinal ganglion cells upon combined co-expression of Math5/Atoh7 and Brn3b/Pou4f2 [474] and the downregulation of Ptbp1 [475] or photoreceptors through two AAV injections, one with beta-catenin to stimulate proliferation and the other with Otx2, Crx, and Nrl [476]. However, the demonstration of the lack of specificity of the alleged glial promoters raised concerns in some of these studies and highlighted the need for proper strategies for tracing the MG as the cell of origin in protocols for MG to neuron reprogramming [477,478,479,480]. Research groups are also searching for alternative or complementary tools for MG reprogramming, such as investigating ways to identify novel cell-type specific regulatory regions to drive gene expression [481], modifications in AAV-carried sequences testing modifications in AAV-carried sequences [482], or screening compounds to increase the neurogenic reprogramming of the MG [483]. A great advance was obtained in the last decades on the identification of the critical approaches necessary to obtain reliable information, which could work as a proof of principle for the investment in regenerative strategies for ocular diseases [484].

However, many challenges are still ahead, as emphasized by alliances between investigators who are working in collaborative networks to promote advances in this field [485]. Even though the relevant data have been accumulated, it is essential to guarantee that translational approaches are designed to promote the generation and integration of new neurons in the retinal tissue to yield the restoration of lost functions.

## 9. Conclusions

The embryonic retina is an invaluable model for analyzing neural structure, function, development, and diseases. This model offers deep insights into the dynamics of neurotransmitter interactions and sheds light on the various factors that influence cellular health within the neural context. Here we updated the role of several key messengers, such as glutamate, GABA, dopamine, the endocannabinoid system, antioxidants, NO, their receptors, and vanilloid channels, among others, which have been summarized. In addition, new gene therapies have been mentioned in this review. Recent advances for glaucoma and diabetic retinopathy have been summarized in this review, which enhances our understanding of retinal disorders and opens up avenues for potential treatments.

## 10. Limitations of the Present Review

We tried our best to cover the major aspects of retinal messengers in the healthy and diseased retinas. The retina expresses many neurotransmitters and gliotransmitters; several neuromodulators were not included (acetylcholine, for instance). The same occurred for signaling pathways (notch, wnt, and others) that were not covered in the present review on retinal physiology, development, or pathological conditions. As it is impossible to address all of them in a single review, the choice was based on the large experience of the senior authors that have been contributing for the last three–four decades. Finally, there are several other retinal diseases alongside the ones that have been discussed. The choice was made based on the prevalence of retinal diseases in humans, including the two most blinding-threatening pathologies in the world.

## Figures and Tables

**Figure 1 ijms-25-01120-f001:**
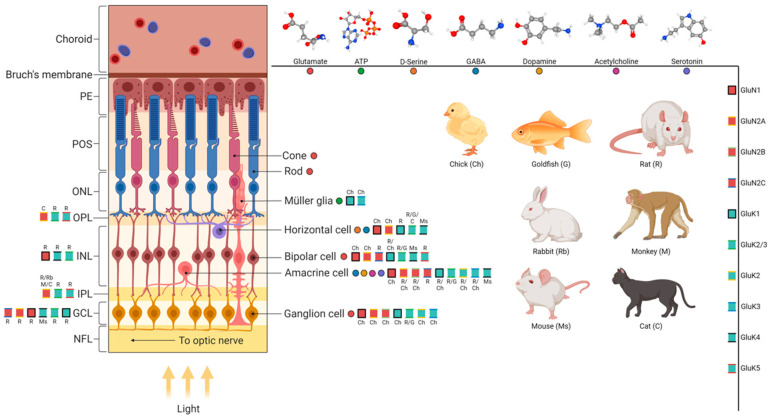
Comparative analysis of the vertebrate retina, selective mediators, and glutamate receptor expression across species. A comprehensive illustration of the retinal structure, neurotransmitter distribution, and glutamate receptor subunit expression across various species. The left side of the image presents a detailed cross-sectional view of the retina, delineating its layered architecture and the cellular components within each layer. The layers are sequentially labeled from the outermost to the innermost as follows: choroid, Bruch’s membrane, pigment epithelium (PE), photoreceptor outer segments (POS), outer nuclear layer (ONL), outer plexiform layer (OPL), inner nuclear layer (INL), inner plexiform layer (IPL), ganglion cell layer (GCL), and nerve fiber layer (NFL). The direction of light entering the retina is indicated by yellow arrows at the bottom, pointing towards the nerve fiber layer. Cell types within the retina are represented by distinct symbols: cones and rods (photoreceptors), Müller glia, horizontal cells, bipolar cells, amacrine cells, and ganglion cells. These symbols are color-coded and positioned to reflect their location within the retinal layers, illustrating the complex interplay of the cells involved in visual processing. The right side of the image features a chart displaying the molecular structures of various neurotransmitters, including glutamate, ATP, D-serine, GABA, dopamine, acetylcholine, and serotonin. It should be noted that acetylcholine and GABA were also present in displaced amacrine cells in the GCL. These neurotransmitters play pivotal roles in retinal signal transduction and are essential for the proper functioning of the visual system. Below the transmitter chart, a key indicates the expression of different glutamate receptor subunits (NR1, NR2A, NR2B, NR2C, GluR5, GluR6/7, GluR6, GluR7, KA1, and KA2) across six species: goldfish (G), rat (R), rabbit (Rb), monkey (M), mouse (Ms), and cat (C). Each species is represented with an icon and a corresponding initial, providing a comparative view of glutamate receptor diversity and its potential impact on visual processing across different vertebrates. Created with BioRender.com—agreement number VL26BHMCYU.

**Figure 2 ijms-25-01120-f002:**
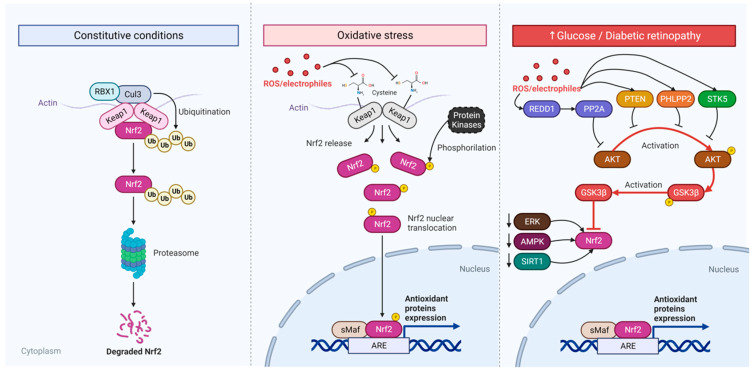
Regulatory pathways involved in retinal stress responses and diabetic retinopathy progression. Cellular mechanisms of the retinal response to stress under constitutive conditions, oxidative stress, and in the context of high glucose (HG) or diabetic retinopathy (DR). The left panel depicts the constitutive degradation pathway of the transcription factor Nrf2, which is bound by the Kelch-like ECH-associated protein 1 (Keap1) and targeted for ubiquitination and subsequent proteasomal degradation under normal conditions. The central panel shows the response to a light to mild oxidative stress, where reactive oxygen species (ROS) or electrophiles modify cysteine residues on Keap1, leading to the release of Nrf2. Nrf2 can also be phosphorylated by protein kinases, which promotes its translocation into the nucleus. Once in the nucleus, Nrf2 binds to antioxidant response elements (AREs) in the DNA, leading to the expression of proteins of the antioxidant response. The right panel focuses on the molecular pathways involved in diabetic retinopathy, a condition characterized by increased glucose levels that lead to retinal damage. Here, the diagram outlines the interplay between ROS/electrophiles and various signaling molecules, including REDD1, PP2A, PTEN, and PHLPP2, all of them are increased by HG/DR, inhibiting AKT. Activated AKT (p-AKT) phosphorylates and inactivates GSK3β, which, in turn, affects Nrf2 activity. Additionally, the diagram indicates the involvement of ERK, AMPK, and SIRT1, which positively regulates Nrf2, but are all decreased in HG/DR, inhibiting the Nrf2-activated antioxidant response. Created with BioRender.com (agreement number: TI268KUA0F).

## Data Availability

All data are available from the corresponding author on request.

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
