# Peer review of "The Healthy and Diseased Retina Seen through Neuron–Glia Interactions"

_ijms, 2024, doi:10.3390/ijms25021120_

Round 1
Reviewer 1 Report
Comments and Suggestions for Authors
In the comprehensive review by Tempon et al., the embryonic retina is highlighted as an invaluable model for analyzing neural structure, function, development, and diseases. This model offers deep insights into the dynamics of neurotransmitter interactions and also sheds light on the various factors that influence cellular health within the neural context. The review is well researched and underscores the significant progress made in genome and transcriptome analyses, which not only enhances our understanding of retinal disorders but also opens avenues for potential treatments.
Author Response
The healthy and diseased retina seen through neuron-glia interactions.
Matheus H. Tempone et al
Review, International Journal of Molecular Sciences MDPI
Manuscript ID: ijms-2810631
Assistant Editor: Maturin Natesungnoen, Ph.D.
Point by Point Response to reviewers.
Reviewer 1
In the comprehensive review by Tempone et al., the embryonic retina is highlighted as an invaluable model for analyzing neural structure, function, development, and diseases. This model offers deep insights into the dynamics of neurotransmitter interactions and also sheds light on the various factors that influence cellular health within the neural context. The review is well researched and underscores the significant progress made in genome and transcriptome analyses, which not only enhances our understanding of retinal disorders but also opens avenues for potential treatments.
Reply: We thank the reviewer comments toward our manuscript
Reviewer 2
The authors present a very deep review retinal cells and its neurotransmitters in vertebrates. A hard work has been done, undoubtedly (30 pages and 490 references). All the current knowledge about glutamate, GABA, dopamine, the endocannabinoid system, and their receptors has been summarized. TRP channels or antioxidants might not be so well-known as neurotransmitters, but they are included, as well. In addition, new gene therapies have been included in the review. Retinitis pigmentosa is one of the inherited retinal dystrophies that has been more studied in this field, and the most recent advances have been well summarized in the review.
The manuscript is well structured. References are adequate. English language is perfect. There is not much left to say or to add. This is the deepest review I have ever read. I congratulate the authors for their hard efforts. This is a valuable review, and all the scientific community will benefit from it.
Reply: We thank the reviewer comments toward our manuscript
Reviewer 3
Esteemed authors,
I congratulate you on the extensive review regarding molecular pathology of the retina.
However, there are some aspects that require your attention:
Regarding the Figures, you need to insert at the end of the captions information regarding copyright.
Reply: We thank the reviewer comments toward our manuscript. Regarding copyright license, information now is provided, as well as a new version of Figure 1 (lines 102-103). In the end of Figures, please find
Figure 1. "... Created with BioRender.com - Agreement number VL26BHMCYU." Figure 2. "... Created with BioRender.com - Agreement number TI268KUA0F."
We are also sending the image license terms for the novel figure 1.
You have an enormous number of abbreviations, please insert at the end of the manuscript a list of abbreviations.
Reply: We thank the reviewer for mentioning the abbreviation list that is now found in lines 1524-1548
At the end of the manuscript insert a subsection regarding the limitations of the present review. For example, this is not a systematic review, and others.
Reply: We have added a paragraph regarding limitations (lines 1523-1532), that now reads… Limitations of the present review: We tried our best to cover the major aspects of retinal messengers in the healthy and diseased retina. The retina expresses many neuro- and gliotransmitters; several neuromodulators were not included (acetylcholine, for instance). The same occurred for signaling pathways (notch, wnt, and others) were not covered in the present review in retinal physiology, development, or pathological conditions. As it is impossible to address all of them in a single review, the choice was based on the large experience of the senior authors that have been contributing to the last 3-4 decades. Fi-nally, there are several other retinal diseases besides the ones discussed. The choice was made based on the prevalence of retinal diseases in humans, including the two most blinding-threatening pathologies in the world.
Moreover, you need to insert a short paragragh of conclusions at the end of the text.
Reply: A short paragraph of conclusions is now found in lines 1512-1521, which reads…
The embryonic retina is an invaluable model for analyzing neural structure, function, development, and diseases. This model offers deep insights into the dynamics of neu-rotransmitter interactions and sheds light on the various factors that influence cellular health within the neural context. Here we updated the role of several key messengers, such as glutamate, GABA, dopamine, the endocannabinoid system, antioxidants, nitric oxide, their receptors and vanilloid channels and others, which have been summarized. In addition, new gene therapies have been mentioned in the review. Recent advances for glaucoma and diabetic retinopathy have been summarized in the review, which enhances our understanding of retinal disorders and opens avenues for potential treatments.
Author contribution at Line 1516 needs to be formatted according to MDPI instructions.
Reply: Now lines 1549-1552 reads: Each author has made substantial contributions to the conception or design of the work; on the acquisition, analysis, or interpretation of data – here we acknowledge Dr Fernando Garcia de Mello, who throughout his career influenced the careers of many generations of researchers in Brazil and abroad.
At line 1523, about Data availability statement. I advise you to use a common expression such as: All data are available from the corresponding author on request.
Reply: Done (line 1557-1558)
You have 490 References, please check for duplicate, also format according to MDPI instructions for authors.
Reply: Done

Reviewer 2 Report
Comments and Suggestions for Authors
The authors present a very deep review retinal cells and its neurotransmitters in vertebrates. A hard work has been done, undoubtely (30 pages and 490 references). All the current knowledge about glutamate, GABA, dopamine, the andocannabioid system, and their receptors has been summarized. TRP channels or antioxidants might not be so well-known as neurotransmitters, but they are included, as well. In addition, new gene therapies have been included in the review. Retinitis pigmentosa is one of the inherited retinal dystrophies that has been more studied in this field, and the most recent advances have been well summarized in the review.
The manuscript is well structured. References are adecquate. English language is perfect.
There is not much left to say or to add. This is the deepest review I have ever read. I congratulate the authors for their hard efforts. This is a valuable review, and all the scientific community will benefit from it.
Author Response

(The authors gave the same response as above.)

Reviewer 3 Report
Comments and Suggestions for Authors
Esteemed authors,
I congratulate you on the extensive review regarding molecular pathology of the retina.
However, there are some aspects that require your attention:
Regarding the Figures, you need to insert at the end of the captions information regarding copyright.
You have an enormous number of abbreviations, please insert at the end of the manuscript a list of abbreviations.
At the end of the manuscript insert a subsection regarding the limitations of the present review. For example this is not a systematic review, and others.
Moreover, you need to insert a short paragraph of Conclusions at the end of the text.
Author contribution at Line 1516 needs to be formatted according to MDPI instructions.
At line 1523, about Data availability statement. I advise you to use a common expression such as: All data are available from the corresponding author on request.
You have 490 References, please check for duplicate, also format according to MDPI instructions for authors.
Looking forward the receive the improved version of your manuscript.
Author Response

(The authors gave the same response as above.)
